



# Drone-based magnetic and multispectral surveys to develop a 3D model for mineral exploration at Qullissat, Disko Island, Greenland

Robert Jackisch[1,†], Björn H. Heincke[2], Robert Zimmermann[1, ††], Erik V. Sørensen[2], Markku Pirttijärvi[3], Moritz Kirsch[1], Heikki Salmirinne[4], Stefanie Lode[5], Urpo Kuronen[6], Richard Gloaguen[1]

[1]Helmholtz Institute Freiberg for Resource Technology, Helmholtz-Zentrum Dresden-Rossendorf, Freiberg, 09599, Germany
[2]Geological Survey of Denmark and Greenland, Copenhagen, 1350, Denmark
[3]Radai Oy, Oulu, 90590, Finland
[4]Geological Survey of Finland, Rovaniemi, 96100, Finland
[5]Norwegian University of Science and Technology, Trondheim, 7031, Norway
[6]Bluejay Mining Plc, London, W1F0JH, England

*Correspondence to*: Robert Jackisch (jackisch.robert@gmail.com)

† now at: Technical University Berlin, Berlin, 10623, Germany
†† now at: G.U.B. Ingenieur AG, Freiberg, 09599, Germany

## Abstract

Mineral exploration in the West Greenland flood basalt province is attractive because of its resemblance to the magmatic sulphide-rich deposit in the Russian Norilsk region, but it is challenged by rugged topography and partly poor exposure for relevant geologic formations. On northern Disko Island, previous exploration efforts have identified rare native iron occurrences and a high potential for Ni-Cu-Co-PGE-Au mineralization. However, Quaternary landslide activity has obliterated rock exposure at many places at lower elevations. To augment prospecting field work under these challenging conditions, we acquire high-resolution magnetic and optical remote sensing data using drones in the Qullissat area. From the data, we generate a detailed 3D model of a mineralized basalt unit, belonging to the Asuk Member (Mb) of the Palaeocene Vaigat formation.

A wide range of legacy data and newly acquired geo- and petrophysical, as well as geochemical-mineralogical measurements form the basis of an integrated geological interpretation of the unoccupied aerial system (UAS) surveys. In this context, magnetic data aims to define the location and the shape of the buried magmatic body, and to estimate if its magnetic properties are indicative for mineralization. High-resolution UAS-based multispectral orthomosaics are used to identify surficial iron staining, which serve as a proxy for outcropping sulphide mineralization. In addition, high-resolution UAS-based digital surface models are created for geomorphological characterisation of the landscape to accurately reveal landslide features.

UAS-based magnetic data suggests that the targeted magmatic unit is characterized by a pattern of distinct positive and negative magnetic anomalies. We apply a 3D magnetization vector inversion model (MVI) on the UAS-based magnetic data





to estimate the magnetic properties and shape of the magmatic body. By means of using constraints in the inversion, (1) optical UAS-based data and legacy drill cores are used to assign significant magnetic properties to areas that are associated with the mineralized Asuk Mb, and (2) the Earth's magnetic and the paleomagnetic field directions are used to evaluate the general magnetization direction in the magmatic units.

Our results indicate that the geometry of the mineralized target can be estimated as a horizontal sheet of constant thickness, and that the magnetization of the unit has a strong remanent component formed during a period of Earth's magnetic field reversal. The magnetization values obtained in the MVI are in a similar range as the measured ones from a drillcore intersecting the targeted unit. Both the magnetics and topography confirm that parts of the target unit were displaced by landslides. We identified several fully detached and presumably rotated blocks in the obtained model. The model highlights magnetic anomalies that correspond to zones of mineralization and is used to identify outcrops for sampling.

Our study demonstrates the potential and efficiency of using multi-sensor high-resolution UAS data to constrain the geometry of partially exposed geological units and assist exploration targeting in difficult, poorly exposed terrain.

# 1 Introduction

The volcanic rocks of Palaeocene age exposed on Disko-Nuussuaq in central-west Greenland form part of the North Atlantic Igneous Province (Larsen et al., 2016). Due to a similar geological setting the Disko-Nuussuaq area is regarded as analogous to the Norilsk-Talnakh Ni-Cu district in the Siberian trap basalt, and thus a highly prospective region for major Ni-Cu-Co-PGE deposits (Lightfoot et al., 1997; Keays and Lightfoot, 2007). Mineral exploration in the onshore parts of the basin at Disko Island and the Nuussuaq Peninsula dates back more than half a century (Pauly, 1958; Bird and Weathers, 1977; Ulff-Møller, 1990) and currently there are 12 active mineral exploration licenses that cover an area of ~10,000 km$^2$ on Disko-Nuussuaq (Government of Greenland, 2021).

Large parts of the Northern Disko region provide good outcrop conditions at the high plateau steep-slopes, whereas the lower slopes near the coast are covered by debris from Quaternary rock falls, landslides, periglacial deposits and solifluction lobes (Pedersen et al., 2017). This incapacitates ground-based mineral exploration mapping efforts, which is further complicated by rugged topography and the Arctic climate.

Here, high-resolution, multi-parameter three-dimensional models are highly useful to resolve detailed structures and develop exploration models. This has traditionally been achieved by combining results from various exploration techniques (Vallée et al., 2011), with magnetics as a prime method (Nabighian et al., 2005). Systematic airborne geophysics (Brethes et al., 2014, 2018), and remote sensing (Bedini, 2011; Bedini and Rasmussen, 2018) has been used in Greenland to create the uniform physical data basis for such modelling.

The recent development of small-scaled UAS equipped with versatile sensors created a powerful tool in spatial mapping (Ren et al., 2019). Magnetic sensors (Gavazzi et al., 2016, 2019; Malehmir et al., 2017; Parshin et al., 2018; Walter et al., 2020; Zheng et al., 2021), as well as multi- and hyperspectral sensors (Kirsch et al., 2018; Jackisch et al., 2019; Booysen et





al., 2020), on drones makes it possible to acquire data inexpensively and with higher resolution as compared to traditional airborne surveys. Magnetic data are suited to map surface and shallow subsurface structures (Le Maire et al., 2020) and are useful to reveal magnetized rock units and the location of sulphides and iron-oxides (Gunn and Dentith, 1997). Integrated high-resolution RGB and image spectroscopy is employed to detect small-scale mineralization traces and can safely guide
ground teams during exploration (Park and Choi, 2020).

Ground-based measurements and rock sampling are typically carried out as part of a mineral exploration campaign to establish and validate relationships between data measured from indirect airborne and UAS-based survey methods, and the field based mineralogical, lithological and structural data. In addition, geomorphologic properties, e.g., the topography of landslides can be incorporated in the geological interpretation. The tracing of mineralized boulders to discover in situ
mineralization (colloquial: boulder hunting) is regarded as an effective exploration tool (Plouffe et al., 2011).

Landslide geohazard monitoring using UAS is an additional source of valuable data. In the Disko-Nuussuaq region, monitoring has received increased attention lately, highlighting the Nuussuaq basin as a risk area (Dahl-Jensen et al., 2004; Svennevig, 2019). Landslide descriptors (e.g., headscarps) are often hard to identify because their characteristic appearance (e.g., fracture patterns) are eroded or overprinted by continuing mass movements.

In this study, we focus on an area south of Qullissat on the northern shore of Disko Island (Fig. 1), which has seen modern exploration since the early 1990's. Legacy data, which was available to the authors of this paper, include airborne magnetic and active electromagnetic (EM) data as well as petrophysical data from six drillholes intersecting a magmatic body (Olshefsky, 1992; Olshefsky and Jerome, 1993, 1994). However, considering the limited thickness of the mineralized unit, and the lithological complexity of the area due to secondary mass movements, the data coverage of the legacy geophysical
surveys is too coarse to develop a 3D exploration model of the area. Additionally, large rotated blocks may occur at coastal zones or are partially buried by talus.

We complement the existing data with newly acquired high-resolution drone, or unoccupied aerial system (UAS) based multi-sensor data. Magnetic measurements were carried out with a fixed-wing UAS (Jackisch et al., 2019, 2020) at low altitude and with dense line spacing to acquire high-resolution magnetic data. In addition, we conducted a high-resolution
UAS-based multispectral and photogrammetry survey in order to create a precise elevation model and to systematically identify locations with increased iron content for mineralization vectoring. UAS-based data were supplemented with ground-based observations, magnetic surveys, handheld spectroscopy and magnetic susceptibility measurements and laboratory petrophysical and mineralogical analysis of rock samples from one legacy drill core from the area.

We link topography, surface mineralogy and the magnetic data to provide both direct and indirect information about
potentially sulphide-enriched targets. In particular, we use the magnetic data in a 3D magnetization vector inversion (MVI) as a means to constrain the shape of the mineralized body and its main magnetization directions and distribution. Finally, results from all UAS and ground-based data are combined in a joint interpretation of the Qullissat area. The interpretation aims (1) to define and pinpoint potential exploration areas, and additionally, (2) to determine where parts of the targeted magmatic unit are displaced by landslides.



## 1.1 Regional geological setting

The volcano-sedimentary Nuussuaq Basin in western Greenland formed as a rift basin in Early Cretaceous time during rifting of the Labrador Sea-Davis Strait area (Henderson et al., 1981; Chalmers et al., 1999; Dam et al., 2009). Because of the Neogene uplift (Japsen et al., 2005; Bonow et al., 2006), parts of the basin are now exposed in the onshore areas of Disko Island and Nuussuaq Peninsula in central West Greenland. The area is made up of Cretaceous to Palaeocene siliciclastic sediments of the Nuussuaq Group (Dam et al., 2009 and references therein) and Palaeogene volcanic rocks of the West Greenland Basalt Group (Pedersen et al., 2018 and references therein; Pedersen et al., 2018 and references therein). On a regional scale, sediments were deposited in a deltaic environment in the eastern part of the basin (sandstones interbedded with mudstones) while deep marine sediments were deposited in the western part. During Late Cretaceous to Early Palaeocene rifting, sediments were block-rotated and eroded prior to the onset of volcanism (Chalmers et al., 1999; Dam et al., 2009).

The volcanism started in a submarine environment within the actively subsiding Nuussuaq Basin. Early eruptive products were hyaloclastites extruded from eruption centres located to the NW of Disko and Nuussuaq. With the volcanic up-built on the seafloor, volcanic islands were formed and over time volcanism became dominantly subaerial.

The Palaeocene volcanic succession is divided into a lower (Vaigat formation; Fm) and an upper formation (Maligât Fm) which makes up the bulk of the volcanic rocks exposed on Disko and Nuussuaq (Fig. 1b). The early volcanism of the Vaigat Fm was dominated by picritic rocks that erupted in three overall volcanic cycles (Larsen and Pedersen, 2009). The picritic rocks formed from melts generated through partial melting in the asthenosphere. The melts subsequently ascended through the crust and erupted at the surface without much interaction while traversing the crust from source to surface. However, throughout the volcanic pile intervals, crustally contaminated siliceous basalts to magnesian andesites occur (Larsen & Pedersen, 2009; Pedersen et al., 2017; Pedersen et al., 2018) indicating that primary magmas at certain times got contaminated in relatively high-level magma chambers. This is evidenced by the occurrence of partly digested shale and sandstone xenoliths in the volcanic rocks (Pedersen, 1977, 1985; Ulff-Møller, 1977).

## 1.2 Economic mineral potential

The economic mineral potential of the West Greenland Basalt Province is recognized as an equivalent of the Norilsk-Talnakh region with potential for major Cu, Ni, Co and PGE deposits (Keays and Lightfoot, 2007; Rosa et al., 2013). Key similarities are a high proportion of high temperature picritic lavas and a significant volume of sediment-contaminated basalts (Lightfoot and Hawkesworth, 1997).

When primary magmas passed through the sediments en-route to the surface, they reacted at various locations with sedimentary successions modifying the chemical composition of the magmas. The rare native (telluric) iron is observed at several places across Disko and Nuussuaq (Ulff-Møller, 1985, 1990). It is commonly suggested that it formed by the reaction of iron present in the magma with carbonaceous sediments (e.g., marine mudstone, deltaic shales, coal seams) resulting in



extremely reducing environments leading to the precipitation of nickel-ferrous minerals and metallic iron (Howarth et al., 2017; Pedersen et al., 2017; Pernet-Fisher et al., 2017).

Under similar conditions, contamination from sulphur-rich sediments is also responsible for the precipitation of Ni, Cu, Co
and PGE as immiscible sulphide droplets within the silicate magma that are scavenged and deposited (Sørensen et al., 2013). At Illukunnguaq, north-eastern of Disko, a 28 t massive Ni-Cu-Co-PGE boulder has been known and investigated since 1870 (Steenstrup 1901). At Hammer Dal, north-western part of Disko Island, a 10 t massive native-iron boulder has been found in Stordal, 1985 (Ulff-Møller 1986). These boulders indicate that processes leading to massive accumulations of iron-oxide- and sulphide-mineralization have occurred. In a dynamic open magmatic system, where a large volume of mafic nickel-rich
melt streams through dykes and sills, huge Ni-Cu-PGE deposits (conduit type nickel deposit) can be formed. A common intrusive geometry found in large igneous provinces (LIPs) are extensive networks of tabular or saucer-shaped sills linked by dykes, which interact with the sedimentary basin (Barnes et al., 2016). A recent geochemical soil survey, e.g., an extended mobile metal ion study (Blue Jay Mining Plc, 2021), provides further indications of economic Ni-Cu-Co-PGE-Au deposits. In addition, significant amount of Au were reported in native iron cumulate within one core sample (drillhole FP94-4-5)
nearby Qullissat (4.8–38.3 g/t Au; Olshefsky and Jerome, 1994), although the Au content was otherwise low and sporadic in further core samples.

Our main target is a sub-horizontal magmatic body that is located near Qullissat, about 20 km south-east of the well-investigated native-iron-bearing Asuk locality (Pedersen, 1985), and 25 km NW of the known Illukunnguaq Ni-Cu dyke (Pauly, 1958). Our target, and the area, was first described as a sill called "Qullissat sill" (Olshefsky and Jerome, 1994;
Pedersen et al., 2017). It is assumed to be part of the Vaigat Fm and quite similar to Asuk Mb in chemical composition. The presence of sulphides, graphite and native iron in the intrusion (Olshefsky et al., 1995), which are all considered to be conductive, complicates an interpretation that is largely based on electromagnetic data. Detailed magnetic data can provide insight into which components are mainly responsible for conductivity anomalies, because graphite is non-magnetic but conductive, while pyrrhotite is quite magnetic (Gunn and Dentith, 1997). The observed native iron might have significant
ferromagnetic properties (Nagata et al., 1970).

### 1.3 Geochronology and magnetic polarity of the basalt members

Previous palaeomagnetic investigations of volcanic strata on Disko and Nuussuaq (Deutsch and Kristjansson, 1974; Athavale and Sharma, 1975; Riisager and Abrahamsen, 1999, and references therein) showed that a geomagnetic pole reversal took place at ~60.92 Ma (magnetochron C27n–C26r; Cande and Kent, 1995). About two-thirds of the lower–middle
Vaigat Fm are normally polarized, but its upper third and the overlying Maligât Fm are reversely polarized.

Of importance to this paper is the Asuk Mb, which formed at a period of reverse polarisation shortly after the C27n–C26 pole reversal (Pedersen et al., 2017). Adjacent field measurements of the remanent magnetic field near the Asuk locality is from a site, located ~25 km NW of Qullissat at sampling altitudes between 365–1450 m. Field declinations (D) between 123–154° and an inclination (I) of about -73° were reported (Athavale and Sharma, 1975). Remanent magnetic





measurements were also reported from age-equivalent rocks of the Naujánguit Mb at Qunnilik on southern Nuussuaq
(Riisager and Abrahamsen, 1999, 2000), about 60 km northwest of Qullissat, where a reverse polarisation with I = -80.7° and
D = 228.1° were measured.

### 1.4 The Qullissat study area

The study area is located near the abandoned coal mining town Qullissat (70.0844 N, 53.0097 W) on the north coast of
Disko island (Fig. 1). The study area is ~6 km long, ~3 km wide and extends from sea level up to the steep Inussuk cliff
(~600–900 m above sea level (asl); Fig. 2b). The lower parts of the study area (up to 600 m asl) are broadly debris and
vegetation covered, affected by mass-movement and thus only show limited outcrops. The general geology of the area
(Fig. 1c) is described on the official geological map and in more detail, in a photogrammetric cross-section covering the area
(Pedersen et al. 2017, Fig. 161, p. 180).

The lower coastal cliffs (< 100 m asl) are made up of Cretaceous sandstones with shale beds and coal seams of the Atane
Fm, while scattered outcrops in the area up to 400 m asl are generally mapped as undifferentiated volcanic rocks or
intrusions of iron and native iron-bearing magnesian andesite of the Asuk Mb (Pedersen et al. 2017). Although originally
described as a sill, recent investigations of the drill cores intersecting the area suggests that the intrusion might be an
extrusive lava flow (pers. communication with Asger Pedersen, Geological Survey of Denmark and Greenland GEUS,
2020).

The area from ~400–600 m asl is characterized by landslide material from the lower Rinks Dal Mb of the Maligât Formation.
The uppermost part of the study area (~600–900 m asl) consists of volcanic rocks of the Maligât Fm (Skarvefjeld Unit) that
form the Inussuk cliff above the Qullissat area. The location of the magmatic plumbing system and the eruption centres for
the Asuk Mb lavas at Qullissat is unknown. However, considering the viscous nature of the Asuk Mb, the andesitic magma
lavas presumably did not spread far from their eruption site.



**Figure 1.** General overview of the study area on northern Disko in West Greenland. (a) The Qullissat study area is located about 125 km NW of the village Ilulissat. (b) Regional geological map from Disko Island with the Palaeogene basaltic Maligât and Vaigat formations, which are emplaced in a Cretaceous sediment basin and incised deeply by glacial erosion. (c) Geological map of the survey site from this study at Qullissat (modified after Pedersen et al., 2013). (d) Oblique overview of the study area based on stitched helicopter-based RGB photographs. The horizontal distance at coast level is about 5 km and the Inussuk plateau is located at ~900 m asl.

## 1.5 Former exploration data

During the last decades, multiple mineral exploration datasets were acquired within the study area (Olshefsky, 1992; Olshefsky and Jerome, 1993, 1994; Olshefsky et al., 1995; Data et al., 2005). Six boreholes were drilled (Olshefsky et al., 1995) by Falconbridge Greenland Ltd. in 1993 and 1994 (see locations in Fig. 3b). Apart from one, all drillholes were located in the western part of the study area at altitudes of ~300–360 m asl and reached downhole depths between 58–270 m. The five western drillholes FP93-4-1, and FP94-4-2 to FP94-4-5) intersect the top of the target unit, whereas only drillhole FP94-4-5 intersects both the top and base of the magmatic body. Drillhole FP94-4-6 (max depth: 143 m) is located at the east





at low altitude close to the coastline and did only intersect sedimentary units, including coal seams and carbonaceous
siltstones. Magnetic susceptibilities were measured in borehole FP94-4-5 at 0.1 m intervals (Olshefsky and Jerome, 1994).
Legacy geophysical data at Qullissat comprise both ground-based surveys, e.g., two crossing magnetotelluric (MT) profiles
(Data et al., 2005) and magnetics (Olshefsky and Jerome, 1994; survey 4A in Fig. 3c) and a local airborne survey, where
both time-domain EM and magnetic (Fig. 3b) data were acquired (Olshefsky and Jerome, 1993). In addition, the Qullissat
area was covered by the regional Aeromag1997 survey (Thorning and Stemp, 1998). However, the survey line spacing
(~1.0 km) was too coarse to be of use in this study (Fig. 3a). Also, the local airborne survey had rather coarse line spacing of
~200–500 m and significant flight heights of ~150 m above ground level (agl). This magnetic data provided only limited
resolution at the given outcrop scale and did not allow the characterization of the magmatic body in any detail (Fig. 3c). Two
mapped conductive anomalies from the airborne EM measurements and ground MT showed potential for promising
exploration targets (Olshefsky and Jerome, 1993, Data et al., 2005).

**2 Methods and materials**

**2.1 Acquisition and processing of UAS-based magnetic data**

We measured the local magnetic field with a digital three-component fluxgate magnetometer located in the tail boom of a
fixed-wing UAS (type: Albatros VT from Radai Ltd, Oulu, Finland). During surveying, the three orthogonal components of
the magnetic field were recorded together with GPS time, position (latitude, longitude and altitude) and barometric pressure
by a data logger (see Appendix A for details). We used the individual magnetic components to compute the total intensity of
the magnetic field and estimate the horizontal in-flight GPS accuracy positioning to be about ± 1 m. After UAS take-off, the
flights were controlled by an autopilot that followed predefined trajectories. A magnetic base station was set up in the field
to correct for the diurnal field variation during measurements.

The total surface coverage of the Qullissat survey area was ~6.8 km$^2$, which we realized in nine flights. Our fixed-wing had a
mean velocity of 58 km/h, resulting in a mean inline sampling of 2.6 m. The separation between the SE-NW directed flight
lines was 40 m (line azimuth is about 27° anticlockwise from north, Fig. 3d and Appendix A). The total length of the flight
lines was ~220 km and total flight time was ~3.7 h. A nominal flight altitude on the path was defined as 40 m above a terrain
topography defined by a digital elevation model (DEM; Dataforsyningen, 2019). However, the real flight altitudes were
larger (mean: 70 m), because the flight path software added a safety margin for altitudes over with strong topography to
avoid steep pitch angles at abrupt slopes.

After basic data processing, an equivalent layer modelling (ELM; Pirttijärvi, 2003; Nakatsuka and Okuma, 2006) was
applied using the RadaiPros software (Radai Oy, Oulu, Finland). Here, we computed the total magnetic intensity on a regular
grid (20 x 20 m) and at a constant altitude of 40 m above the ground. More details about the ELM method and other
processing steps are provided in Appendix A.



The processed magnetic anomaly map is presented in Fig. 3d. The noise level in the final magnetic data was estimated from
the low-pass filtered corrected data using a standard deviation of the 4[th] difference, which ranges between ~1–5 nT in the raw
magnetic data. For the low-pass filtered (wavelength > 20 m) corrected magnetic data, the corresponding standard deviation
is less than 0.03 nT.

From the total magnetic anomaly map of the UAS-based magnetic data, we calculated a modified magnetic anomaly

upward-continued by 60 m to an elevation of 100 m agl (UP100), the first vertical derivative (VD), the analytic signal (AS)
and the tilt derivative (TLD), all shown in Fig. 4 (Nabighian, 1972; Miller and Singh, 1994; Isles and Rankin, 2013; Dentith
and Mudge, 2014). The combination of those filters helped us to delineate anomaly borders, provide information on local
magnetization strength and increase visual interpretation of both the near-surface and deeper features.

**2.2 Acquisition and processing of fixed-wing multispectral and photogrammetric data, and additional image sources**

Multispectral data was acquired using a SenseFly ebeePlus UAS featuring a Parrot Sequoia multispectral (MSI) camera (1.2
Mpixel) with 4-channels in the VNIR spectral range. Spectral channels (i.e., bands) are centred at 550 nm (green), 660 nm
(red), 735 nm (red edge) and 790 nm (NIR), respectively. The bands are sensitive to chlorophyll related absorptions but also
suited for the detection of iron-related spectral features (e.g., see Jackisch et al., 2019; Flores et al., 2021). In particular the
ratios of 735/790 nm or 660/550 nm are proven useful to map the iron absorption feature associated with Fe-alteration

minerals (Rowan and Mars, 2003; Rowan et al., 2005). UAS-based images were processed by means of structure-from
motion multi-view-stereo photogrammetry using Agisoft Metashape (version 1.6; details in Appendix B), following the
protocols set by various authors (e.g., James and Robson, 2014; James et al., 2016, 2019). The resulting colour-infrared
(CIR) orthophoto (Fig. 2a) and the digital surface model (DSM; Fig. 2b) largely overlaps the area covered by the UAS-based
magnetic survey (Fig. 1c).

The multispectral orthomosaics cover an area of 13 km$^2$ and reveal surface mineral information from spectral absorption
features as well as in morphology, topography and structural features such as landslides visible in the associated DSM. Slope
and topographic position index (TPI after Weiss, 2001) are useful tools to analyse landforms and enhance morphological
formations, for example valleys, slopes, dikes and crests. We used a TPI image to enhance the image contrast of the UAS-
based slope map (Fig. 8b). It is plotted as a semi-transparent mask onto a true colour-composite from a PlanetScope satellite

image (scene id: 20190829-151652-20-1064, Planet Team, 2021) to enhance the coarser-scaled geologic surface
interpretation outside of our UAS survey area (Figs. 13a, 14).

The image mosaics contain cast shadows and strongly varying illumination conditions, therefore we masked most under-
illuminated parts manually. Required topographic corrections of multispectral mosaics were performed using the Mephysto
toolbox (Jakob et al., 2017). The vegetation index (Kriegler et al., 1969) and a band ratio ($\frac{band\ 3}{band\ 4}$, 'simple iron ratio') were

computed and noise got reduced by applying a median filter (kernel size 5 x 5 pixel). To increase the interpretation fertility
of the iron band ratio, we applied a contour algorithm (GDAL/OGR contributors, 2021) on the ratio image to generate





vectorized isolines, using 0.1 ratio-interval steps. This step size enables a connected interpretation at smoother image contrast.

An off-the-shelf DJI Mavic Pro (12.3 Mpixel RGB camera) was used for backup, and to document sampled areas in video

and photography. We mapped one specific basalt outcrop near the coastline that featured numerous rock samples in nadir images to create an RGB orthomosaic (Fig. 12e).

Finally, vessel-based digital single-lens reflex (DSLR) photographs we acquired while sailing from Qullissat town south to the delta area (Fig. 13d, e). The DSLR images fill gaps in the UAS-image coverage along the coal mine area and provide an oblique viewing angle onto the outcropping sediment packages at sea level (Figs. 13d, e). In areas not covered by

photogrammetric UAS-based data, elevation information was supplemented from the ArcticDEM (Porter et al., 2018).

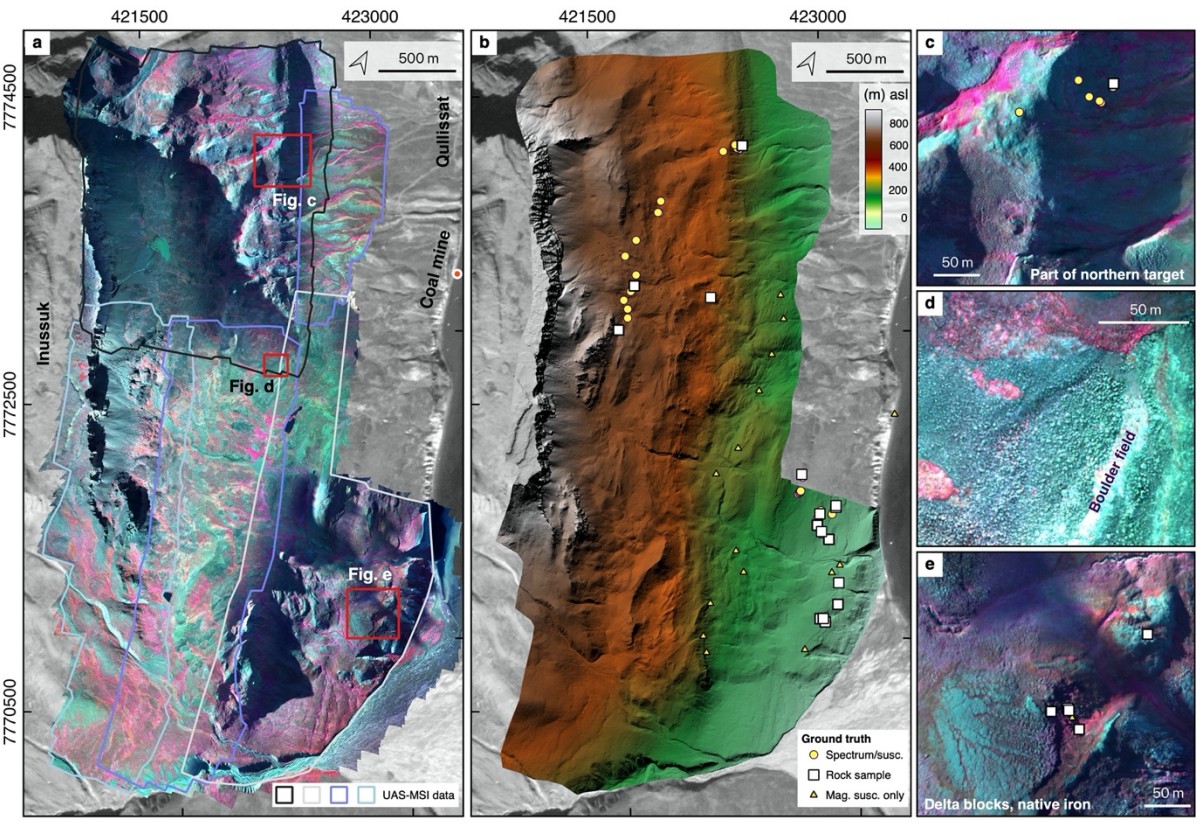

**Figure 2.** Primary data of the multispectral UAS-based surveys after basic processing. (a) Multispectral mosaic at ~ 20 cm GSD in false colour RGB bands 3, 2, 1. The different polygons outline the survey areas of individual flights. (b) DSM having ~36 cm GSD. Locations of collected ground truth data are indicated with symbols. Inset maps enhance view resolution of areas that were ground-sampled during

the study, such as (c) an outcrop associated to the northern part of the target magmatic (d), the central boulder field (e) and outcropping slid volcanic rocks in the southern part.



### 2.3 Ground-based and laboratory measurements

We conducted ground-based measurements such as magnetic surveys, susceptibility and spectroscopy measurements for validation. In addition, magnetic and electric properties were measured on drill core FP94-4-5 samples, together with a
qualitative mineralogic analysis using scanning electron microscopy (SEM).

### 2.4 Ground magnetic surveys

Ground-based magnetic measurements were done at two different areas at Qullissat (survey 4B and 4C in Fig. 3c) with a GEMsystems GSM-19 Overhauser magnetometer at a resolution of 0.01 nT. Measurements of the total magnetic field were made with a mean inline sampling of 1.12 m and 1.49 m and line spacings of 50 m and 100 m, for the northern and southern
survey, respectively. Time and positions were obtained by an integrated GPS receiver and were internally stored together with the magnetic data. A standard data processing for ground-based magnetic measurements was performed with Geosoft Oasis Montaj from Seequent. Diurnal variations in the total magnetic field were removed from all ground magnetic measurements using data from an observatory at Qeqertarsuaq (Godhavn station, identifier: GDH), located at a distance of ~90 km at southern Disko Island.

**2.5 Magnetic susceptibility measurements, handheld spectroscopy and grab sampling**

We collected representative grab samples and conducted magnetic susceptibility (Fig. 10a, b) as well as handheld spectroscopic measurements exclusively on basaltic rocks at Qullissat. Magnetic susceptibilities were measured with a KT-10v2 magnetic susceptibility meter. For the majority of locations, we used the average of 3-5 measurements.

Ground spectra were recorded in the VNIR-SWIR range (400–2500 nm) featuring a spectral resolution of 3.5 nm (1.5 nm
sampling interval) in VNIR and 7 nm (2.5 nm sampling interval) in the SWIR. Radiance values were converted to reflectance using a pre-calibrated PTFE panel (Zenith polymer) with >99% reflectance in the VNIR and >95% in the SWIR range. Each spectral record consists of 10 consecutive measurements. We performed a recalibration after 20–50 scans each, to account for instrument drift. Around 3–5 measurements per GPS point were taken. The main areas covered with susceptibility and spectroscopy are the northern part of the magmatic body, a flat-lying outcrop near the shoreline sediments
and at selected spots near a river delta in the south of the investigation area (Figs. 1c, d).

### 2.6 Petrophysical and scanning electron microscope measurements on core samples from drill core FP94-4-5

The core from the legacy drillhole FP94-4-5 (location in Fig. 3b) is stored in the drill core archive of GEUS (Geological Survey of Denmark and Greenland) and was accessible for this study. We selected core samples in ~10 m intervals in a depth range from 49.5–215.7 m, which comprised samples from both the magmatic body and the sediments above and
below. On these 19 samples, a variety of petrophysical properties were measured at the petrophysical lab of GTK (Geological Survey of Finland) in Espoo. These measurements include magnetic properties such as the induced and natural



remanent magnetization (NRM) as well as the inclination and declination of the remanence, electric properties (not shown here) as the resistivity and the chargeability (both in time and frequency domain), and the dry bulk density. Samples had a diameter of ~3.5 cm and lengths between ~5–10 cm. Susceptibility and NRM is measured with an AC susceptibility bridge
(Puranen and Puranen, 1977) and a fluxgate magnetometer (Airo and Säävuori, 2013), respectively. Petrophysical measurements were complemented with detailed mineralogical SEM analyses to link physical characteristics with specific components such as native iron, pyrrhotite, graphite and magnetite.

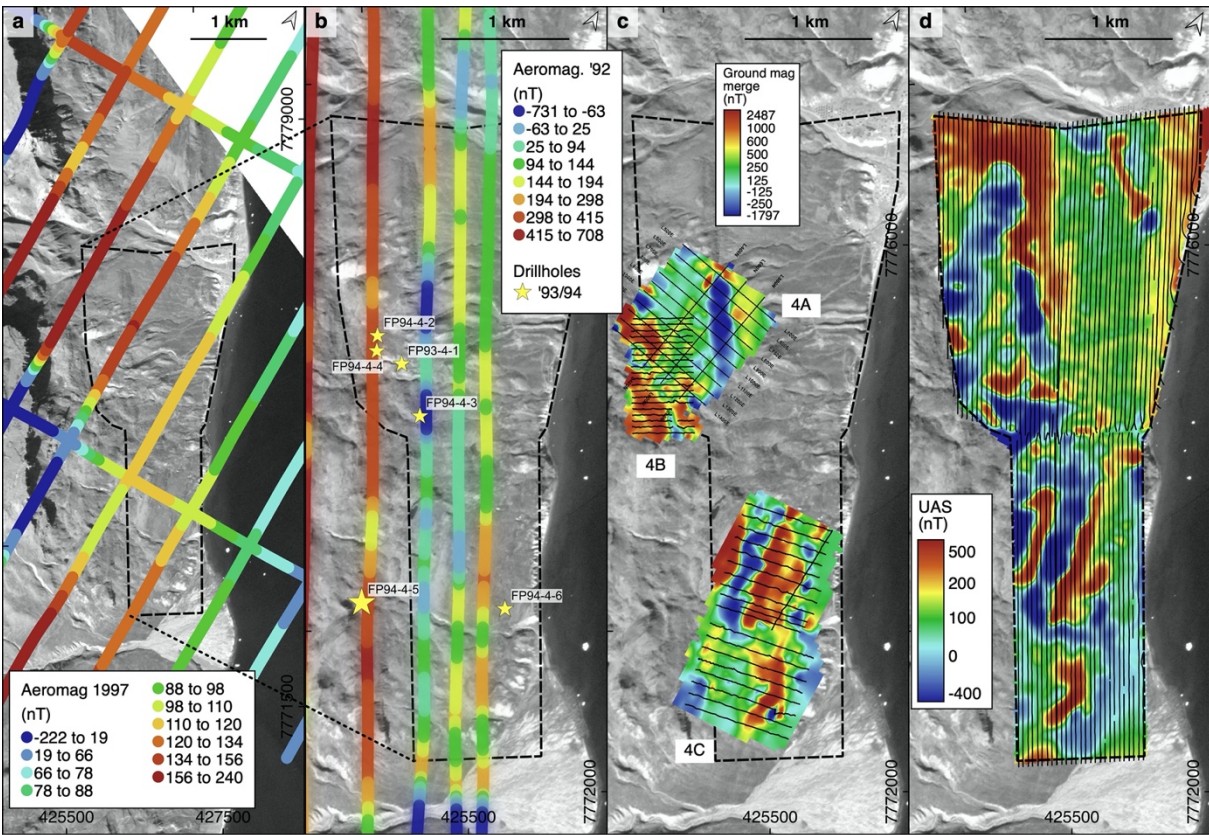

**Figure 3.** Overview of legacy airborne geophysical, ground-based magnetic and UAS-based magnetic data with the dashed line
highlighting the UAS-based magnetic survey area of this study. The residual magnetic anomalies are shown (a) from the regional AEROMAG97 survey (Thorning and Stemp, 1998), (b) from a local exploration airborne survey (Olshefsky and Jerome, 1994) and (c) from ground-based surveys, conducted by Falconbridge Ltd. (4A; Olshefsky and Jerome, 1995), and collected during our field campaign (4B and 4C). The UAS-based magnetic anomaly map is shown in (d) for comparison. The flight trajectories are shown as black lines, and positions of legacy drillholes are indicated in (b) as stars.

**3 Results**

**3.1 Magnetic analysis**

Aeromagnetic data are considered as crucial to improve understanding of exploration targets (e.g., Ni-Cu-PGE) in terms of size and depth, and, under favourable circumstances, even age if advanced petrophysical measurements such as thermal





demagnetization are conducted (Austin and Crawford, 2019). The main characteristics of our observed anomalies, detected
by UAS-based magnetics, are summarized in Table 1 and briefly described below.

Most of the magnetic anomalies are located in the western and central part of the study area at elevation > 200 m asl. They
are arranged in a complex pattern that is trending in a NW–SE direction (strike of ~320º to 325º; Fig. 3d). Many of these
anomalies have short wavelengths and have both distinct high and low amplitudes in the local residual magnetic field (-400
to -50 nT; Fig. 3d ) and in its VD (-17 to -5 nT/m; Fig. 4b). Since the AS shows high values for both the high and low-value
anomalies (Fig. 4c), it indicates that sharp gradients between magnetic highs and lows exist (Nabighian, 1972; Roest
et al., 1992).

Two major short wavelength anomalies (A and B in Fig. 4) are located in the eastern part of the survey area. The positive
anomaly A is located directly at the shoreline in the central part of the survey area close to the old coal mine. The positive
anomaly B, located close to the Qullissat village, is elongated and strikes NW–SE. At its south-eastern end, a dipole-shaped
anomaly is present.

In the northern and southern part, some short wavelength anomalies (anomalies C and D) show elongated shape (see Figs. 3d
and 4b, c, d). The anomaly C in the northern part is oriented in NW–SE (strike ~325$^{0}$), but the direction of the southern
anomaly pattern D, which consists of a negative anomaly that is margined by a positive anomaly at both sides, is oriented
more towards N–S direction (strike ~355$^{0}$). In the central area, the anomalies are randomly distributed, and a preferential
strike direction is not observed (see pattern G in Fig. 3d and Figs. 4b, c, and in a larger area as chaotic patterns in the TLD in
Fig. 4d). Several of these features (C, D, E, F and G) are also observed in the ground magnetic surveys (Fig. 3c) which
confirms the reliability of anomalies identified from the UAS-based magnetic data.

These short wavelength anomalies disappear in the upward continued version (UP100) of the residual magnetic anomaly
(Fig. 4a). Instead, negative anomalies become more pronounced in the central western part (see anomalies C, D and G in
Fig. 4a); while further to the north and south, the anomalies tend to be positive in the UP100 (see anomalies E and F in the
south; Fig. 4a).





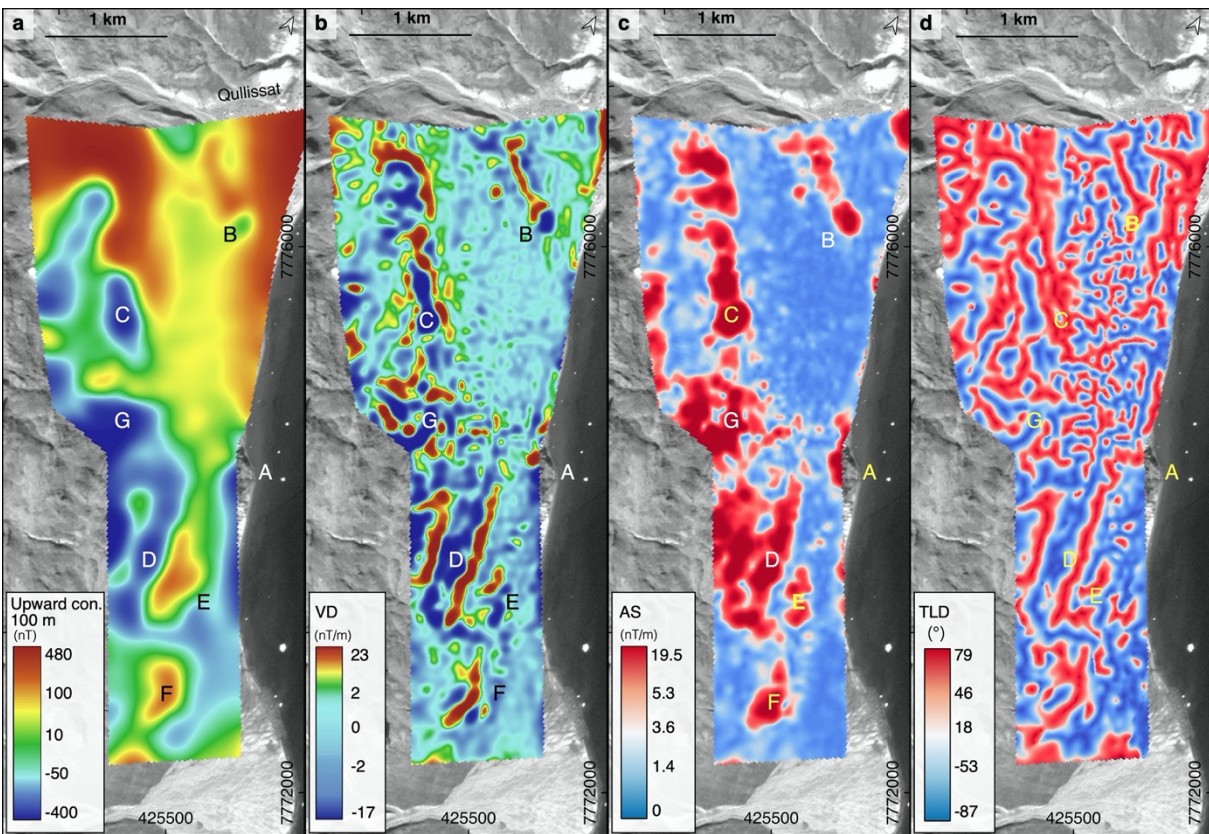

**Figure 4.** Magnetic filter maps obtained from the residual magnetic anomaly of the UAS-based magnetic data in the Qullissat area. (a) The residual magnetic anomaly is upward continued (upward con.) to 100 m (UP100) to enhance anomalies at larger depths. (b) The first vertical derivative (VD) is presented to enhance short-wavelength near-surface features. (c) The analytic signal (AS) amplitude is presented to highlight areas with increased magnetization, independent of the magnetization direction. (d) The tilt derivative (TLD) map highlights both surficial and deeper structural trends.

Table 1. Magnetic features identified in the residual magnetic anomaly (Fig. 3d) and with associated derivates (Fig. 4).

| No. | Description of magnetic features |
| --- | --- |
| A | Positive feature in magnetic anomaly that disappears in the UP100; high values in the AS; isolated anomaly that is associated with the 'Nunngarut block' |
| B | Southern end is dipole shaped in magnetic anomaly map; clearly visible in AS |
| C | Pronounced negative anomaly in magnetic anomaly map with high negative values also in VD and UP100; high AS values; strike towards north (328º); associated with magmatic Asuk Mb |
| D | High negative anomaly in residual field, with negative values in VD and UP100; high AS values; strike 348º N observed in TLD; associated with magmatic Asuk Mb, "main block" |
| E | Positive anomaly in magnetic anomaly map that is also visible in its UP100, VD and AS; located nearby sedimentary outcrops along the coast; iron observed in the field; associated with the "coastal blocks" |
| F | Large positive anomaly in residual field (> 500 nT) near delta; isolated anomaly with high AS values; observed iron in the field; associated with the "delta blocks" |





| | |
|---|---|
| G | Chaotic pattern of magnetic anomalies that is observed in the VD and TLD; in contrast, a pronounced negative magnetic anomaly is present in the UP100 of the magnetic anomaly |

## 3.2 3D magnetic modelling

A 3D magnetization model of the Qullissat area was developed from the fixed-wing UAS-based magnetic data using a deterministic Magnetization Vector Inversion (MVI). Employing such magnetic inversion, that accounts for the full magnetization vector became more common recently (Ellis et al., 2012; MacLeod and Ellis, 2013, 2016; Liu et al., 2017; Li et al., 2021) and has been applied e.g., in the mapping of complex volcanic domains (Miller et al., 2020). UAS-based magnetics with close flight line spacing and low ground clearance are especially suited to measure magnetic remanence (Dering et al., 2019) and experiments confirm that they are reasonably sensitive (Cunningham et al., 2018) to indicate remanent contributions of magnetizations (Calou and Munschy, 2020).

We have chosen an MVI approach because borehole measurements show that the remanent component of the magnetization partly dominates the investigated magmatic body (see section petrophysical properties). Under such circumstances scalar magnetic inversion only considering the induced magnetization component may generate misleading results. However, MVI suffers from a higher non-uniqueness such that additional information as from e.g., core logs, measured petrophysical properties, surface structures and different lithologies need to be incorporated to produce geologically plausible models that are consistent with other geoscience data and observations. Therefore, geologically relevant information was added stepwise as constraints during the inversion process.

The general inversion setup as described in Ellis et al (2012) is:

$$\min \phi\,(\boldsymbol{m}) =\ \phi_D(\boldsymbol{m}) +\ \lambda\phi_M(\boldsymbol{m}) \tag{1}$$

$$\phi_D(\boldsymbol{m}) =\ \sum_{j=1}^{M}\left|\frac{F(\boldsymbol{m}_j) - d_j}{e_j}\right|^2 \tag{2}$$

$$\phi_M(\boldsymbol{m}) =\ \phi_{M,Smooth}(\boldsymbol{m}) + \phi_{M,Ref}(m) = \sum_{p=1}^{3}\sum_{y=1}^{3}\left|w_{y,p}\partial_y\widetilde{m}_p\right|^2 + \sum_{p=1}^{3}\left|w_{0,p}\cdot\left(\widetilde{m}_p\ -\ \widetilde{m}_{0,p}\right)^T\right|^2 \tag{3}$$

$$\lambda:\ \phi_D(\boldsymbol{m}) =\ \chi_T^2 \tag{4}$$

with $\phi$ being the objective function to be minimized, $m = (\widetilde{\boldsymbol{m}}_1, \widetilde{\boldsymbol{m}}_2, \widetilde{\boldsymbol{m}}_3) = (m_{1,1}, \dots, m_{1,N}, m_{2,1}, \dots, m_{2,N}, m_{3,1}, \dots, m_{3,N})$ being the model vector containing the three components $p = (1,2,3)$ of the magnetization of all voxels $k = (1, \dots, N)$, $d$ being the observed data vector of total magnetic field anomaly at each measuring point $j = (1, \dots, M)$ and $e$ being their





associated data errors. The resulting magnetizations are given in susceptibility equivalences and have SI units. After the targeted error weighted data misfit was reached in an inversion, the data term $\phi_D$ and regularization term $\phi_M$ were balanced in the objective function relative to each other. A solution with the highest regularization was found, i.e., the largest regularization parameter $\lambda$ was selected, where the targeted chi-squared data misfit $\chi_T^2$ was reached (for details see Ellis et al. 2012).

In all presented inversion tests, a smoothing constraint associated with $\phi_{M,Smooth}$ was added as regularization and an iterative reweighting inversion focus (Portniaguine and Zhdanov, 2002) option, sharpening anomalies in the model, was active. The smoothing term had, in all inversion tests, weights $w_{y,p}$ of 1 in all directions $y$ and for all components $p$. In addition, the inversion was constrained towards a reference model $m_o$ associated with the term $\phi_{M,Ref}$ for some of the inversion runs.

The main part of the 3D model covered by magnetic data from the UAS survey was discretized in $200 \times 267 \times 71$ cells in x, y and z-directions surrounded by a background model with stepwise increasing cell sizes. In the main part, cell sizes in x and y directions were 20 m, whereas cell sizes in the z direction increased with depths from 10 m at 425 m asl down to 108 m size at -1094 m asl. As the surface topography, we used the regional digital surface model.

The ELM processed UAS total magnetic anomaly data at a constant altitude of 40 m were used as input $d$. An error of 5 nT

was assumed for all data points and accounts for inaccuracies in the instrumentation and positioning as well as for high frequency component loss during the ELM processing.

In the first run, no geological information was used in constraints (i.e., no term $\phi_{M,Ref}$ was added). The target misfit was reached in a few iterations, that was also the case for all follow-up runs. However, the first unconstrained run resulted in a geologically unrealistic model (not presented here), where strong magnetic anomalies were partly located in areas associated

with the non-magnetic sediments both below and above the magmatic body.

In the following, the different geological units and their magnetic properties were considered in the inversion by establishing constraints of the type $\phi_{M,Ref}$. For this, the shape of the tabular magmatic body and the location of the basalts along the Inussuk cliff face were estimated. The upper surface of the magmatic body was constructed by interpolation of drillhole intersections of the five Falconbridge drillholes (FP93-4-1 to FP94-4-5), and from outcrop exposures observed in

multispectral data (ratios and topography), and RGB images. The top of the outcrops exposures were extracted from the UAS-based DEM (Figs. 5a, 2b). Only the deepest drillhole FP94-4-5 intersected the base of the magmatic body (Olshefsky et al., 1995). To estimate the base, we considered the difference of the top (263 m asl) and base (131 m asl) of the magmatic body in this drillhole as the general thickness (132 m) and downward-shifted the top surface with this value. Afterwards, this base surface estimate was compared with the mapped geology along the surface. At locations, where the base surface

intersects outcropping sediments and basalt, it was modified by shifting it upward and downward, respectively (Fig. 5a). The foot of the basalt cliff (Maligât Fm) was partly covered by debris, but at outcropping sections the surface was observed as



horizontal. Therefore, the associated upper boundary surface of the model environment was considered as a flat and horizontal plane at 400 m asl (Fig. 5a).

After adding the magmatic units into the model (Fig. 5b), the remaining part of the model is assumed to be associated with

non-magnetic sedimentary units, which is in agreement with field observations. To consider this information in the MVI, we set up a reference model with zero magnetization for all voxels and in all three directions ($m_o = 0$). For the voxels associated with non-magnetic sedimentary rocks, the corresponding parameter weights $w_{o,p\,(sediments)}$ were set to 0.5 for $p = 1, 2, 3$, but for voxels containing the magmatic units, the weights $w_{0,p\,(basalt)}$ were all set to 0.0 ensuring that only the sediment areas were constrained towards small magnetic values.

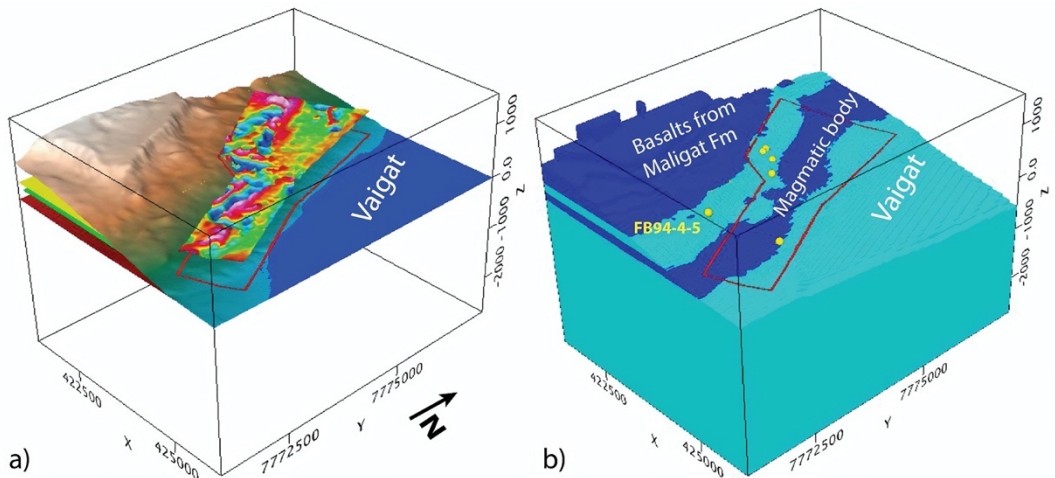

**Figure 5.** 3D voxel-based model for the magnetic inversion. (a) Topography from the regional DEM is shown together with layers associated with the base of the basalts of the cliff (yellow), and with the top (green) and base (red) of the magmatic body. In addition, the map of the total magnetic field from the fixed-wing drone survey is presented, whose data were used as input in the inversion. (b)
Discretized model used in the inversion. Cells associated with the magmatic body and the basalts from the cliff wall (both in dark blue
colours) were derived from the layers presented in (a) and were differently constrained in the inversion as the remaining model (see description of the constraints in the inversion). Yellow dots indicate the locations of the drillholes and the red polygon outlines the area covered by the fixed-wing drone survey.

Inversion results are presented in Figs. 6 and 7. Only the central part of the model is displayed that is covered by drone-based data, since the remaining areas are less well-resolved. Higher magnetization values > 0.01 SI are almost solely placed in

areas defined as the magmatic units and particularly in the mineralized body. The absolute values of the magnetization are with a few exceptions not larger than 0.1 SI (maximum value: ~0.158 SI). In the eastern part, there are only two minor anomalies, which are not located within these units and marked with A and B (Fig. 6b), and no higher magnetizations are assigned to depths below the magmatic body. Within the body the resulting distribution of the magnetization direction is complex and, dependent on the anomaly, high magnetization values are observed for all three components in x- y- and z-

directions. The z-component of the magnetization shows both positive and negative values for different anomalies within the magmatic body (Fig. 6b, c, d). Despite the complexity, the shapes of many of the anomalies show a preferred orientation in a N–S to NNW–SSE direction (see Fig. 7d–i).





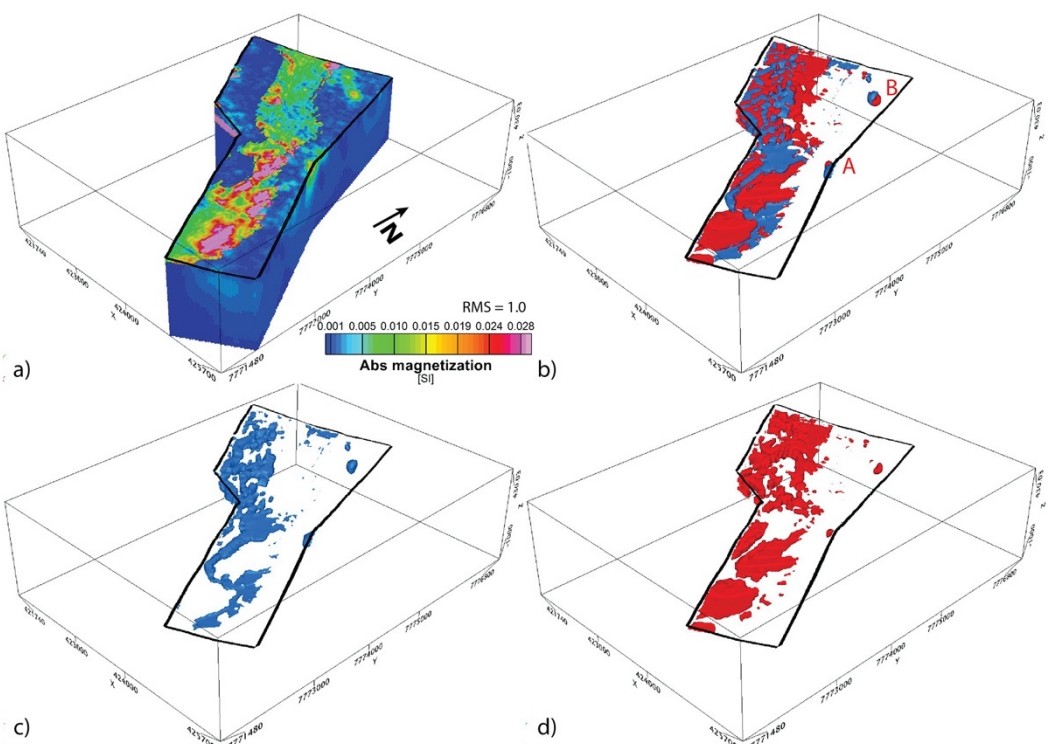

**Figure 6.** Results from the MVI test, where cells associated with sediment units were constrained towards a non-magnetic reference
model, while cells associated with magmatic rocks remained unconstrained. Only the shallow central part of the model down to a depth of
1000 m is shown which was covered by data from the fixed wing survey (black polygon). (a) The final magnetization distribution is
presented as absolute values of the magnetization vectors. (b) Only cells with absolute magnetization values > 0.01 SI are shown as iso-
surfaces. Blue and red colours are associated with locations where the z-component of the magnetization points out of the ground (z-
component is positive) and into the ground (negative z-component), respectively. These two contributions are presented separately in (c)
and (d).

In a last run, the impact of the magnetic field direction in the inversion was also considered. It was assumed that the Earth's

magnetic field and the palaeomagnetic field at the formation of the Asuk Mb were oriented antiparallel and constrained by

the direction of the ongoing magnetic field (i.e., parallel if the induced part of magnetization dominates and antiparallel if the

remanent part dominates). Our assumptions, model constraints and the resulting inversion model are summarized in

Appendix C. The resulting model shows some artificially appearing patterns of small-scaled anomalies (Figs. C1 and C2 in

Appendix C).



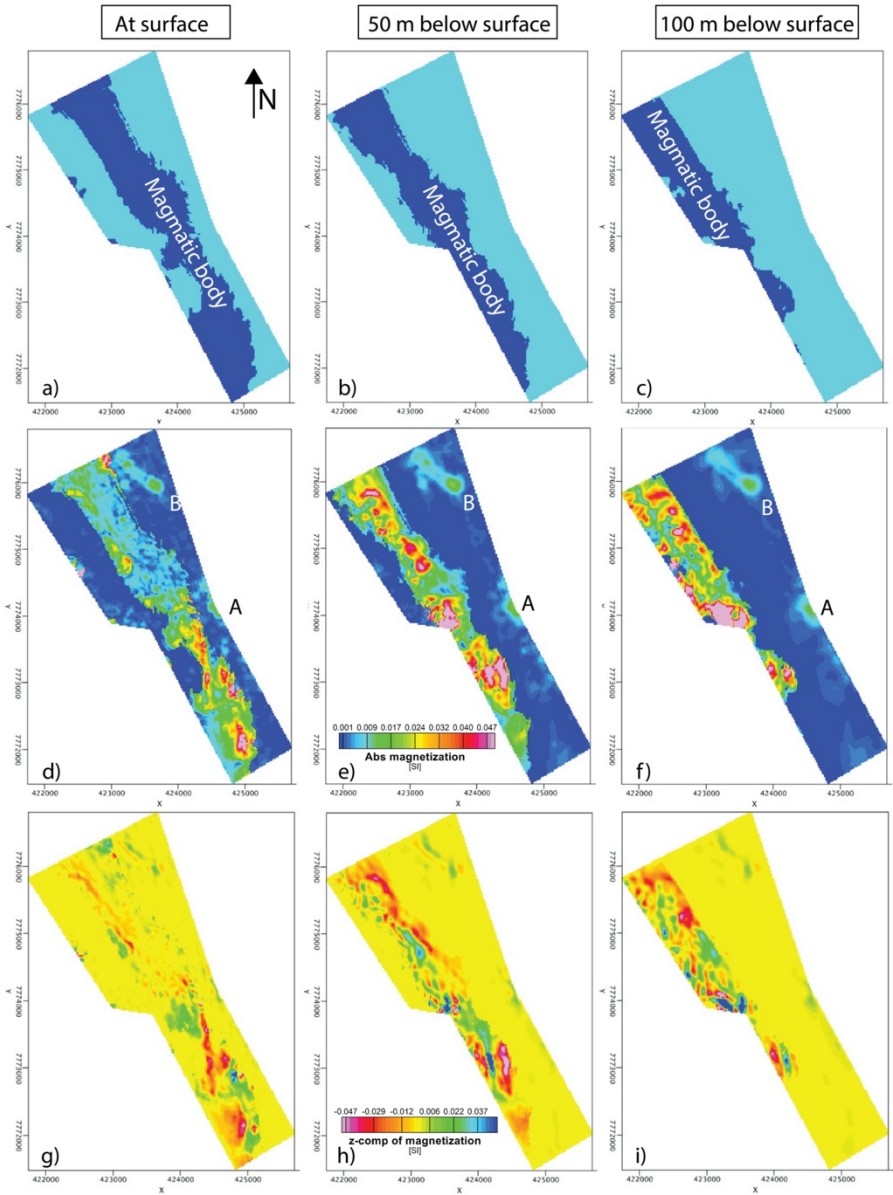

**Figure 7.** Three depth slices through the resulting inversion model, where cells associated with sediment units were constrained towards a non-magnetic reference model, while cells associated with magmatic rocks remained unconstrained. Results are considered at the surface (1st column), and 50 m (2nd column) and 100 m (3rd column) below the surface topography. In (a) to (c) the reference model is shown, where parts associated with target magmatic rocks are shown in dark blue colours. In (d) to (f) and (g) to (i), the absolute value and the z-components of the magnetization are shown, respectively.

### 3.3 Observations from UAS-multispectral and photogrammetry data

Our multispectral surveys cover the whole region of interest and in addition the cliff of Inussuk with GSDs between 0.18–0.36 m. (Fig. 2). We show the presence of iron-bearing outcrops by means of mineralization proxies for iron alteration





minerals and reveal landslide features, e.g., scarps and lobes. The vegetation index (NDVI) mapping (Fig. 8a) illustrates the distribution of widespread low-lying arctic vegetation that covers the surface. NDVI values range between 0–0.71, and we consider pixels with NDVI > 0.3 as dominated by vegetation and are masked out in the image analysis. Vegetation occurs mainly in gently sloping areas and in proximity to water sources, e.g., near stream beds and minor water pathways, all the

way up till below the Inussuk plateau. The iron-sensitive band ratio ($\frac{band\ 3}{band\ 4}$) shows values > 1.0 for ~8% (1.1 km$^2$) for the vegetation-masked orthomosaics. Areas with elevated iron-ratios are distributed across the whole study area (see central area in Fig. 12b). Small blocks and outcrops can be identified by elevated iron band ratios. Large clusters have surface areas of 40,000–90,000 m$^2$ and are located below the boulder field (Fig. 12b).

The high-resolution UAS-based DSM (Fig. 2b), a hillshade (not shown), the slope (total gradient; Fig. 8b) and a TPI

(Fig. 12c) map were used to identify landslide-related features within the study area. A prominent headscarp is visible in the DSM over a length of 1.2 km at an altitude level between 320–350 m asl (Fig. 8d), and is identified by its concave shape in the contour lines. Smaller rockslides and landslide blocks, which are visible in the DSM (Figs. 8d and 12), are dominant in regions at elevations above 200 m and coincide with the general rockslide area (Fig. 1c). Some of the slid blocks showed glacial abrasion during field examination.

The slope of the topography in the Qullissat area generally rises from the shoreline (slope 0–5°) towards the foot of the cliff (slope 15–40°), where the area is undulating and affected the UAS-based magnetic flight altitude. Outcrops form numerous terraces and the slope maximizes at the exposed cliffs of Inussuk (slope > 75°). Most outcrops identified near the coastline (< 250 m asl) have lobate forms, appear strongly disintegrated and are oriented approximately parallel to the shoreline. This trend is observed in the iron band ratio map, where clusters with higher ratios are often arranged in stripes parallel to the

shore (Fig. 12b). The largest outcrop has a size of ~500 × 200 m (150–220 m asl) and its location coincides with the negative values of the magnetic anomaly D (Figs. 4a, 8d, 13a 'main block').



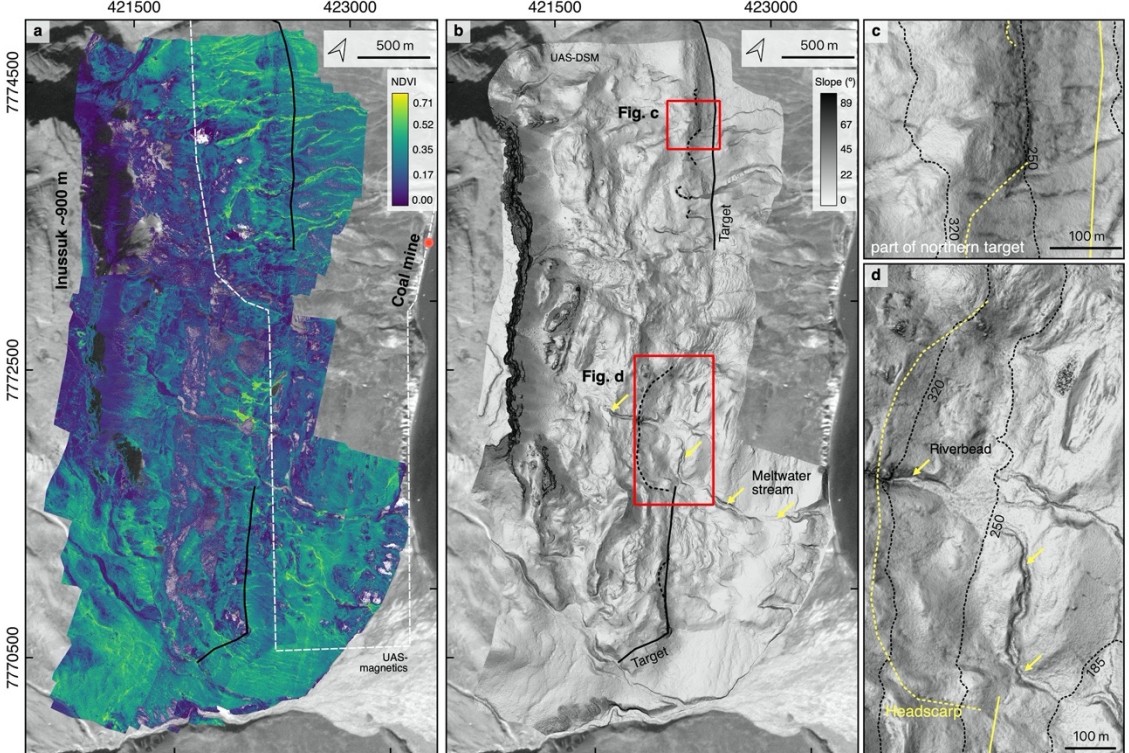

**Figure 8.** Images derived from the UAS multispectral and photogrammetry data. (a) NDVI mosaic derived from the Sequoia camera scenes depicts vegetation occurrence. (b) Slope map (in degree) illustrates the amplitude of the topographic gradient. Inset maps show (c) close-up of the northern part from the targeted magmatic (sampled) and (d) detached blocks with the interpreted headscarp boundary, and a deeply incised meltwater stream.

## 3.4 Ground-based spectroscopy and magnetic susceptibility

The characteristic iron-absorption feature between 850–930 nm (Hunt and Ashley, 1979; Crowley et al., 2003) is pronounced in spectra of most observed magmatic rock samples (Fig. 9). At the same outcrops, we observed small staining of orange-yellow and reddish to black-shaded alteration minerals, for example goethite-hematite, yellow–orange jarosite or limonite along outcrops (Fig. 9a). A colour transition from blackish-lustrous to red on some outcrop surfaces is expressed by a change in the surface spectral response. Streak tests on samples from these locations showed a reddish-brown to dark-ochre colour and their spectra showed a slight absorption band shift from 663 nm towards 671 nm.

Spectral features of other mineral types were not observed on the surfaces of magmatic rocks, but an abundance of lichen-related absorptions is visible in most spectra as absorption pattern in the short-wave infrared region between 1730–2100 nm. These patterns are often caused by the hydroxyl group and can be characteristic for the presence of lichen, which are abundant in arctic environments (Salehi et al., 2017).

We measured magnetic susceptibility exclusively on magmatic rocks. The susceptibilities are relatively high in the study area (mean value of 0.025 SI and maximum value of ~0.01 SI, Fig. 10a) and are within the range of around $10^{-3}$ to 0.01 SI,





typically observed for basalts (Clark and Emerson, 1991). There is no major trend of magnetic susceptibility values with their sampling locations (see Fig. 10b, c), although higher surface values > 0.03 SI were only measured on less weathered rock surfaces from the basaltic outcrops in the south eastern and northern parts, which are mapped as in situ outcrops of the target magmatic body (Pedersen et al., 2017). Measurements on rocks with iron-stained alterations, and on rocks located above the target magmatic body (Fig. 10c, presumably Maligât Fm) all had susceptibility values < 0.03 SI.

An area thoroughly investigated with ground spectroscopy and susceptibility measurements is an outcrop near the coast (Fig. 9a and the close-up Figs. 12e, 13e), named "coastal block" in the following. It is located 100 m asl at 300 m east of the main landslide block. Spectra show a pronounced $Fe^{3+}$ absorption (Fig. 9a) and magnetic susceptibility range from 0.03–0.07 (Fig. 10a). This outcrop coincides with the positive magnetic anomaly E in the drone-borne magnetic data and high iron ratios indicate iron abundance here (Figs. 12b, e).

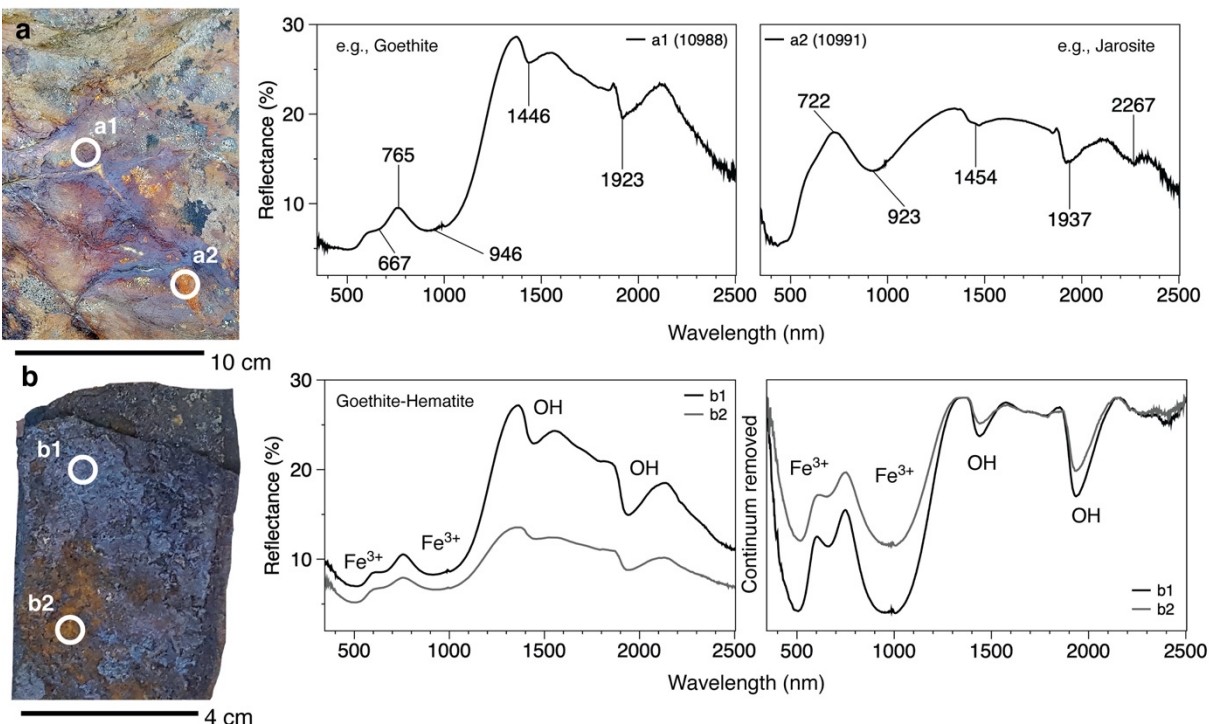


**Figure 9.** Spectral measurements taken on a basaltic outcrop, which correlates with the magnetic anomaly E in Fig. 4 with further remote sensing and magnetic characteristics shown in Fig. 12 a, b (see close-up location in Fig. 12e). (a) The rock surface shows iron-staining and absorption patterns, typically for iron oxide-hydroxide (plotted absorption positions taken from Crowley et al., 2003). (b) Spectra (b1, b2) from a sample (GEUS567321) which was scanned under laboratory conditions. Reflectance spectra (left plot) and continuum removed 515   spectra (right plot) highlight the Fe and the OH⁻ related-absorption features.

### 3.5 Petrophysical properties from cores of drillhole FP94-4-5

The location of drillhole FP94-4-5 was selected on the basis of conductivity anomalies in airborne EM data, a ground-magnetic low and anomalously high gold assays (Olshefsky and Jerome, 1994; Olshefsky et al., 1995). It was the only





drilling that intersected the whole magmatic body at Qullissat and was probed for Ni, Cu and sulphides. Since the drill cores
were unoriented and the actual dip of the borehole was not measured, but assumed to be vertical down to its maximum depth
of 270.5 m (Olshefsky et al., 1995), we consider the measured inclination of the magnetization as rather imprecise.
Therefore, the inclination is only used qualitatively and carefully in further interpretation. We mainly focus on the results
from the density, susceptibility and remanent magnetization measurements (Fig. 10d).

Core logs show the presence of carbonaceous sediments and sandstones in the upper part of the borehole (depth down
hole: 0–50.2 m), before the magmatic body was intersected. The first metres of the body (depth: 50.2–58.1 m) are described
by Olshefsky et al. (1995) as a volcaniclastic breccia, which possibly represents a taxite (~54.80 m), but the remaining part
of the body consists of fine-grained mafic rocks (depth: 58.1–190.5 m). Below the body, rocks comprise carbonaceous
siltstone, shale and sandstone and several thin coal seams (depth: 190.5–270.5).

The magmatic body shows significantly different petrophysical behaviour in its upper (downhole depth < 127 m) and lower
parts (depth > 127 m; Fig. 10d; 127 m downhole depth corresponds to 190 m asl; asterisks marks the contact of the two
parts). In the upper part, the densities are systematically higher (2774–2803 kg/m$^3$), but magnetic susceptibilities (0.001–
0.04 SI), remanent magnetization (0.2–2.4 A/m) and Königsberger ratio Q (0.1–0.7) are smaller than in the lower part
(densities: 2761–2774 kg/m$^3$, susceptibility: 0.08–0.12 SI, magnetization: 5–15 A/m, Q: 1.4–2.7). The change in the
petrological parameters is abrupt at the transition (or contact) and a single sample at this depth shows reduced densities
(2725 kg/m$^3$). However, no obvious change in texture and composition was observed during visual inspection.

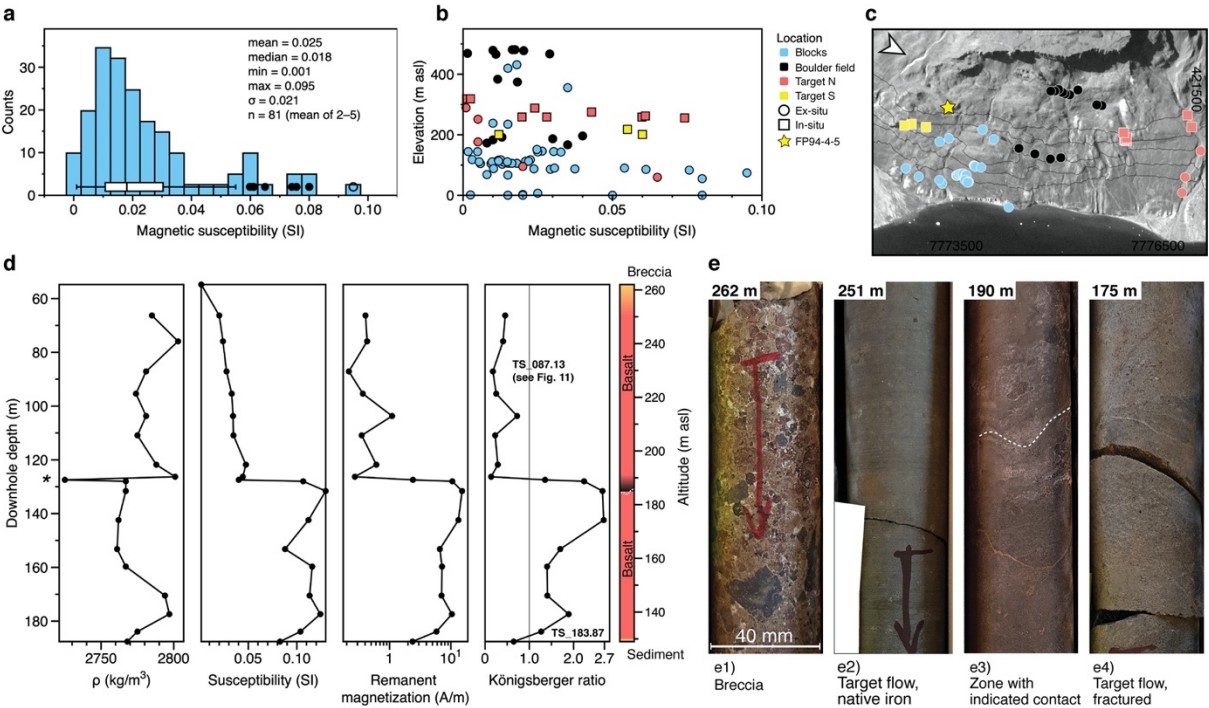

**Figure 10.** Ground-based susceptibility measurements and petrophysical logs from the drillhole FP94-4-5, based on our measurements. (a)
Magnetic susceptibility distribution from handheld measurements. (b) Magnetic susceptibility is plotted against altitude (in m asl) with a





location-based colour scheme. The locations are given in map (c). (d) Petrophysical measurements of density, magnetic susceptibility, remanent magnetization and computed Königsberger ratio for 19 core samples from the drillhole FP94-4-5. Geologic description taken from Olshefsky et al. (1994), altitude in m asl given for comparison. The asterisk marks the contact between the upper and lower part of the magmatic body, having different petrophysical properties. (e) Photographs of four representative core sections (m asl).

Our mineralogical re-investigations of the legacy cores show that the fine to medium crystalline basaltic flows above and below the contact consist predominantly of plagioclase, orthopyroxene, minor clinopyroxene, and very minor reliclike

olivine with an insertal matrix of K-bearing and Fe-bearing glass phases. Native iron is predominantly preserved in the upper block (Fig. 11a, c–e), whereas magnetite and Cu-sulphides, such as chalcopyrite and cubanite are more enriched in the lower block (Fig. 11b, f–h). Nickel-iron phosphides, including schreibersite, and pyrrhotite, are present in both blocks with moderately higher quantities in the upper block. Graphite is present in the upper and lower block without any significant difference. Native iron occurs as larger sub-rounded to irregular shaped blebs mostly within tens of micrometres but up to a

few hundred micrometres in size in the upper block. Native iron commonly shows a rim of magnetite and/or a low-density Fe-oxide alteration phase (Fig. 11d). Frequently, a 'cleaner' appearing secondary native Fe phase is forming a thin rim around the Fe-oxide phase. Furthermore, native Fe is present as micron-sized droplets within the matrix and pheno- and xenocrysts. Pyrrhotite commonly coats the native Fe and Fe-oxide phases and is often associated with graphite. Pentlandite flames and chalcopyrite (and minor cubanite) occur within pyrrhotite (Fig. 11f). In the upper block, the amorphous phase

hisingerite coats the graphite flakes (Fig. 11d). NiFe-phosphides occur in schreibersite composition, but also in more Ni-rich undefined NiP phases (Fig. 11d, f). The higher occurrences of magnetite in the lower flow correlates with high magnetic susceptibilities and remanence in the petrophysical measurements (Fig. 10d).







**Figure 11.** (a) Photograph of a polished thin section of a fine crystalline basalt sample from the upper flow (087.13 m core depth); (b) Photograph of a polished thin section of a fine to medium crystalline basalt sample from the lower flow (183.87 m core depth); (c) to (d) Backscatter electron micrographs (BSE) of sample at 087.13 m with preserved native iron [Fe] blebs and Fe-droplets [Fe-dr] and minor magnetite (Mag) and Fe-oxide (Fe-ox) in association with pyrrhotite [Po], schreibersite (Scb) and graphite (Gr) and hisingerite (His). Brt = barite, Di = diopside, Opx = orthopyroxene; (e) Reflected light micrograph of the same sample with larger native Fe blebs with a magnetite rim and association with pyrrhotite and smaller micron-sized Fe-droplets dispersed in matrix; (f) Reflected light micrograph of sample at 183.87 m core depth with no preserved native Fe, but magnetite in association with pyrrhotite. Pyrrhotite with pentlandite [Pn] flames and chalcopyrite inclusions; (g) to (h) Backscatter electron micrographs (BSE) of the same sample with pyrrhotite in association with magnetite and graphite and siderite (Sd)-filled fractures that also cross-cut a glass (Gl)-filled amygdule. Fe-Mg-chl = Fe-Mg-chlorite.







## 4 Interpretation and discussion

The integration of UAS-based high resolution magnetic, spectral and photogrammetric data allows a more detailed interpretation of the targeted magmatic body (Fig. 12), than previously possible. Magnetic anomaly maps and derivates together with the constrained MVI model enable us to extend the information into the subsurface to propose a reasonable estimate of the extent and shape of the body. Further interpretation is presented in 3D views (Fig. 14).

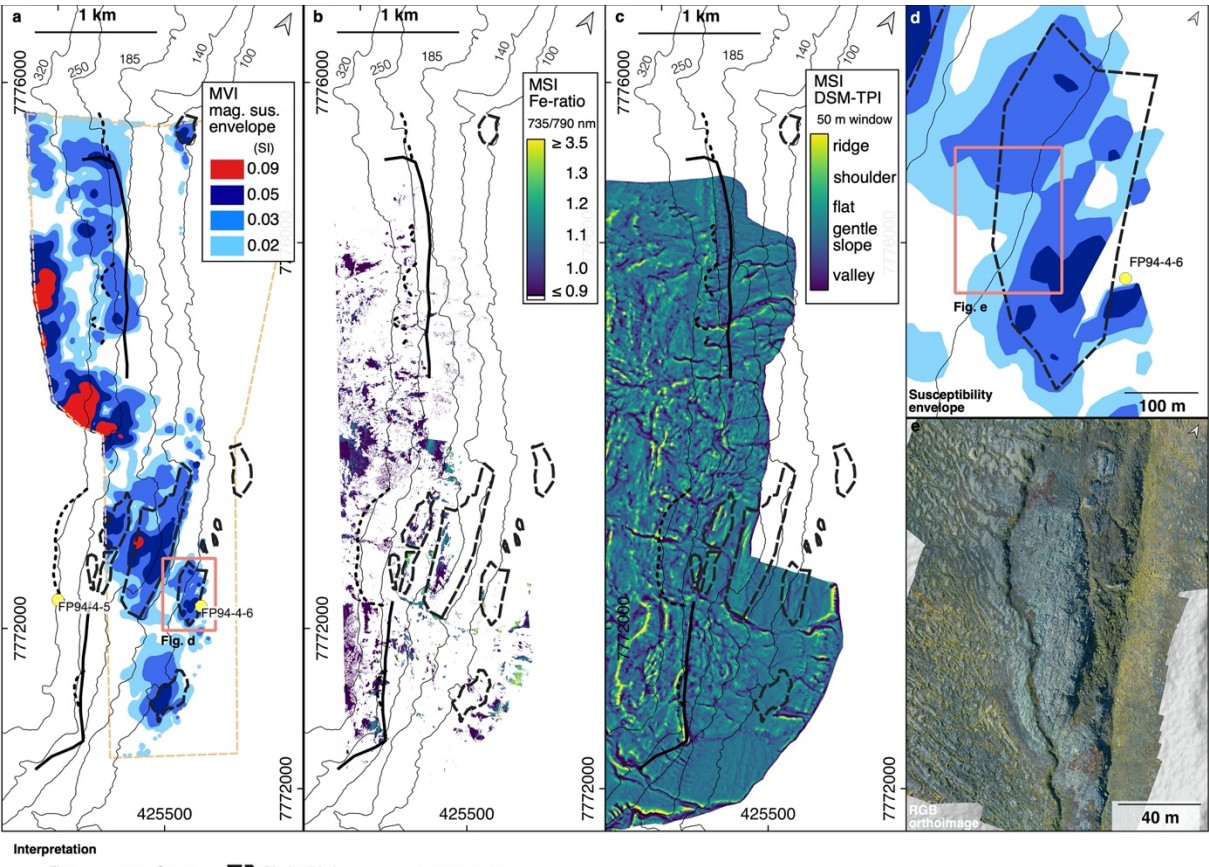

**Figure 12.** Integration of UAS-based data. (a) Isosurfaces of magnetization amplitudes obtained from the constrained MVI inversion. (b) Iron band ratio from multispectral UAS mosaics showing abundance of iron-rich alteration products on the surface (c) Photogrammetry-based TPI calculated from pixels in a 50 x 50 m moving window shows graduation from incised valleys, flat slopes, to ridges. (d) Isosurfaces of magnetization amplitudes from the MVI model, (e) RGB orthomosaic from a single RGB reconnaissance flight (DJI Mavic).

**4.1 Location, shape and size of the mineralized body**

Based on the distributions of (1) outcrops of magmatic rocks identified from the DSM and (2) distinct magnetic anomalies (Figs. 3d and 4a–c), we located the targeted magmatic unit between ~140–320 m asl (Fig. 14b). However, due to landslide activities in the central and southern part of the survey area, ~3 km south of Qullissat, several detached blocks slid down and we identified ex situ basalt blocks down to the sea level (Figs. 12, 13, 14). We estimated the surface exposure of the main





target body with ~7.5 km² in the study area, detached and slid blocks from the body cover an area of ~0.56 km² (Fig. 13b).
       Since magnetic anomalies are observed at the boundaries of the UAS-based magnetic survey, we assume that the body
       continues beyond the survey area below the mountain range towards NW and W. However, it would be difficult to identify
       the magmatic body by magnetic measurements there, because the magnetization of younger basaltic units (mainly Maligât
       Fm) at higher elevation obscure the magnetic response of the body. Towards the S and E, the erosion elevation level is lower

than in the central area and the magmatic body is eroded.
       We interpret the targeted body to be a flat-lying, tabular-shaped body, from information of mapped outcrops in the DSM and
       from the available drill hole elevation data (Olshefsky and Jerome, 1994; Olshefsky et al., 1995). This model is in agreement
       with the results from the constrained 3D magnetic inversion, because almost all magnetic anomalies can be modelled with a
       body having magnetic properties within a reasonable data range (susceptibility equivalences of 0.1–0.15 SI). However, the

limited resolution of the magnetic method does not allow us to reliably interpret further details of shape and thickness
       variation (Blakely, 1995). The appearance of pronounced long-wavelength negative magnetic anomalies in the upward
       continued version of the residual anomaly (see anomalies D and G in Fig. 4a) can be explained by magnetizations with a
       dominating reverse polarization that are located at considerable depths and, hence, supports that the body has a significant
       thickness of ~130 m.

The alternation of high-frequency anomalies in the UAS-magnetic data with strongly varying intensities (i.e., the high
       analytic signal of feature D; Figs. 4b, c, 13c) support our understanding that low magnetic values over the body are not
       created by lack of magnetic material. More likely, there is a significant remanent magnetization contribution that is oriented
       in a distinctly different direction than the induced magnetization. This is also supported by the results from the constrained
       inversion. There are no indications that further major magnetic material e.g., from intrusions are located underneath this

body (see results from the constrained inversion). This also means that a feeder structure of the magmatic body is probably
       not located within the survey area.
       There are plausible explanations why the anomalies A and B (Figs. 3, 4, 6, 7) are located outside of the estimated body.
       Anomaly A, the "Nunngarut" block, is in the central-coast area that is most affected by mass movements and may be
       associated with a larger fragment of the body that slid downward (see next section about landslide features). However,

anomaly A also coincides with the location of the entrance from the former coal mine shaft, where some metallic mining
       equipment has been left. Anomaly B is located within the Qullissat village and may be associated with a construction built
       on a solid rock foundation that is described as native-iron bearing. Anomaly C is described as part of the magmatic target
       unit both in the regional geologic map (Pedersen et al., 2013), as well as the photogrammetric cross section of the Qullissat
       area in Pedersen et al., 2017. Other anomalies D, E, and F (Fig. 13a, b) are associated with displaced material which

locations are only roughly sketched in the cross-section (Pedersen et al., 2017) . Handheld rock samples from those blocks
       (e.g., Figs. 9b, 10c, 12e) contained native iron, pyrrhotite and magnetite, observed on thin-sections (sample location in
       Fig. 2c, e). Those samples also showed a similar texture as seen in the cores of the drill hole FP94-4-5 (Fig. 11).





## 4.2 Linking landslide features to the exploration target

The curved shape of the head escarpment of the landslide identified in the UAS-DSM proposes a concave rupture surface
(Figs. 13a, b and 14) and following known landslide classification systems (Varnes, 1958; Hungr et al., 2014), it can be
considered as a rotational rock slide. The material from the landslide is uniformly composed of mafic rocks and consists of
one large block ("main block" in Fig. 13a) and a number of smaller blocks (as the "delta block", "coastal block" and
"Nunngarut block" in Fig. 13a). Most of the slid material occurs in proximal distance to the head scarp ("main block") and
seems to be moved only slightly, but some larger rotated blocks (e.g., "delta block", "coastal block"; Fig. 13a, e) were
identified at distances of up to ~1 km from the headscarp (interpreted from multispectral maps and DEM data in Figs. 8b,
12c). Several of these blocks were associated in former investigations as parts of the targeted Mg-rich andesite intrusion
from the Asuk Mb (Pedersen et al., 2017).

The magnetic anomalies A, D and E (Fig. 4), and probably the patterns F and G, are located within areas that are affected by
landslide movements, but the anomaly C is located in a presumably stable area. The anomaly pattern D is associated with the
largest "main block" and its deviation of the strike directions (strike ~355°) compared to the general NW-SE arrangement of
magnetic anomalies and the strike direction of anomaly C can be explained by a rotating component (along a virtual z-axis)
during the landslide event. Blocks D and E are surrounded by streams, where flows carved into the more brittle rock
fragments and buried crevices.

The locations of the "coastal block" and "delta block" coincide with a magnetic anomaly (E and F, Fig. 4) indicating that
these landslide blocks consist of magmatic rocks with elevated magnetic properties proposing that they also originate from
the targeted magmatic body. Samples taken near the slid "delta block" and from the northern part of the intrusion, which is
not affected by landslide movements (Figs. 8c, 13a, e), are similar in their geochemical and microscopic compositions
(Fig. 11) strengthening this interpretation. Also, the Nunngarut block located immediately at the shoreline and adjacent to the
coal mine coincides with a small, but distinct positive magnetic anomaly in the residual magnetic anomaly (Fig. 3d). This is
observed in the analytic signal (feature A in Fig. 4c) and results in a spot with elevated magnetizations in the MVI models
(feature A in Figs. 6 and 7). Rocks of this block have a similar geochemical composition as the intrusion (Olshefsky and
Jerome, 1995) and the sample AF0903 from the Nunngarut block (Olshefsky and Jerome, 1993) and the GEUS sample
156690 (Pedersen et al., 2017), which is taken from the southern part of the targeted magmatic body (Fig. 14). Those
samples show similar contents of Mg (6.5–7.6%) and Fe (10–12%) (Pedersen et al., 2017). The chaotic anomaly pattern G in
the central-upper part of the survey area may indicate disrupted rocks from landslide movements. However, this area is
covered with breccia and hyaloclastites, as well as talus material from the cliffs and the magnetic response can also be
explained by other deposition processes.





### 4.3 Distribution of iron alteration products

UAS-based multispectral iron ratios provide information on the distribution of iron-bearing minerals as a proxy for
mineralization. We anticipated that the use of these data for interpretation is limited because spectral identification of
surficial iron occurrence relies on one band ratio combining two bands with broad spectral ranges and low sensitivities. In
addition, the ratios were affected by cast shadows in multispectral images. Especially band 4 (790 nm) has a higher
uncertainty due to low reflectance of volcanic rocks. Therefore, the ratios can be systematically biased and particularly
multispectral pixels with higher ratios (> 3.0) might be unreasonably high. In similar studies, we recommend further
examination, if high ratios occur as spatially isolated anomalies. Even after data cleaning, numerous pixels with non-
illuminated edges and shaded zones remain in the orthomosaic. Therefore, we re-evaluated selected spectral absorption zones
by visual interpretation, by using the UAS-based false-colour orthomosaic (Fig. 2a) and additional RGB imagery from high-
resolution satellite images (Team Planet, 2021). This auxiliary information helped to remove obviously false absorption
zones, e.g., shadowed sedimentary units.

Iron ratios > 1.0 pinpoint outcrops with iron alteration and values > 2.0 are interpreted as patches that can indicate potentially
mineralized blocks and boulders (in total 0.02 km$^2$ or roughly 3–5% of the covered surface). We considered those zones of
interest for closer ground inspection (Figs. 2, 12b). Many of the spots with high ratios in the upper western part can be
associated with basalt blocks and talus material that originate from the adjacent Inussuk mountain, where material from the
lava flows and hyaloclastites of the iron-bearing Skarvefjeld Unit (Maligât Fm) were emplaced. In the lower central and
eastern part, high ratios are likely associated with iron from the magmatic body of the Asuk Mb. The largest landslide block
("main block") expresses ~50,000 m$^2$ of measurable iron absorption (iron ratio > 1.05, Fig. 12b). Also two other outcrops
("delta block" and "coastal block") in the southeast of the study area had surfaces with elevated band ratio values. They were
closely investigated from the ground and iron stains were found (Fig. 12b; "coastal block" outcrop is captured by the DJI
Mavic RGB images in Fig. 12e). Our handheld evaluation spectroscopy indicated a subtle change from an iron-oxide
(hematite) to a hydroxide (goethite; specific spectral features Crowley et al., 2003), a crystal structure change, which is
related to a compositional alteration.

### 4.4 Mineralogical considerations and explanations for magnetic anomalies

The occurrence of goethite ($\alpha$-FeOOH) and hematite ($Fe_2O_3$) is observed as specific absorption features in the surface
spectra at five investigated outcrops and are associated with iron-bearing magmatic bodies (Fig. 9). These oxy-hydroxides
can both be alteration products of the magnetite and titanomagnetite abundant in the basalt (Pedersen et al., 2017), or the
result of corrosion of native iron (Figs. 10, 11). Magnetite, titanomagnetite and native iron are ferrimagnetic (magnetic
susceptibility ~1−100 SI for magnetite; estimates for native iron in nature are limited due to its rare occurrence) and all can
contribute considerably to the magnetic behaviour in the contaminated magmatic body. Another contributor to the magnetic
response could be monoclinic pyrrhotite ($Fe_{1-x}S$; Clark, 1997; Austin and Crawford, 2019).





First results of our re-investigation of drillcore FP94-4-5 with SEM and petrophysical measurements indicate that the higher
         magnetic properties (Q, susceptibility, remanence) in the lower part of the borehole (Fig. 10d) correlate with a higher content
         of magnetite. In contrast, native iron is present in significant amounts only in the upper part having lower magnetic
         properties, which suggests that native iron is not the main source for magnetic characteristics. Since the content of pyrrhotite
         is in the same range or lower than the content of magnetite and, because the magnetic ferro-properties are distinctly higher
than from pyrrhotite, we concluded that the magnetization from magnetite is more dominating than from pyrrhotite.
         Moreover, the Königsberger ratios from monoclinic pyrrhotite in rocks are typically very high (Q ~10–100's, depending on
         its domain state; see Fig. 3 in Clark, 1997), but our observed Q values do not exceed 2.7 in the core samples (Fig. 10d). This
         implies that the magnetic responses are probably not diagnostic for the mineralization in this body, which has major
         implications for the use of magnetic data for sulphide exploration in this area.

Although native iron is not the main source for the magnetization in the body, it is harder to judge if its contribution is
         significant in its upper part, which magnetization is still significant although clearly smaller than in the lower part (Fig. 10d).
         Nagata et al (1970), who investigated lunar material, stated that very fine-grained native iron can have a superparamagnetic
         character. This means that its ferromagnetic remanence can have a major component that changes its orientation in direction
         of the outer field (viscous magnetic remanence) at temperatures significantly below its Curie temperature (771 $^{o}$C). If the
native iron has such a superparamagnetic character at Qullissat, much of its viscous component may be turned along the
         current Earth magnetic field direction resulting in lower Q values. Indeed, we observe lower Q values in the upper part (Q
         = 0.1–0.7) of core FP94-4-5 where native iron is abundant, than in the lower part (Q = 1.4–2.7). However, other factors as
         e.g., grain sizes of magnetite have an impact on the magnetic domains and finer grains are usually associated with larger Q-
         ratio (Dunlop and Argyle, 1997).

The abrupt change in all measured petrophysical properties in the drillhole at a depth of 127 m is notable (Fig. 10d). The
         question arises if the change is associated with (1) two separate successive magmatic pulses, which differ in their
         geochemical and accordingly also petrophysical properties, or (2) with segregation processes in the liquid magma during
         formation. Several observations propose the former explanation as abrupt change in mineralogy and magnetic properties at
         the contact zone. Also the low-density spike ($\varrho$ = 2725 kg/m$^3$) at this depth could be explained as a zone of increased
weathering by exposure to the ground surface or alteration by heat from a subsequent magmatic event.
         If these two units are from separate magmatic events, they could be extrusive lava flows intruding into the soft sediments or
         covered by material afterwards, and not a sill, as stated in former studies (Olshefsky and Jerome, 1994). Currently, a
         manuscript from Pedersen et al. is in preparation that will discuss this aspect in detail. Higher values in both induced and
         remanent magnetization in the lower part of drillhole FP94-4-5 (Fig. 4a) suggest that lower flows of the first pulse are more
magnetic than flows of the later pulse. This would explain why distinct anomalies can be found in the upward continued
         version of the residual magnetic anomaly (Fig. 4a, anomalies D and G), where stronger magnetizations at larger depths are
         enhanced. The negative character of these anomalies is in agreement with higher Königsberger ratios in the lower part of the
         drillhole (Q > 1.0) provided that the body was emplaced during a period of magnetic pole reversal. Accordingly, the





appearance of such negative anomalies is in accordance with the interpretation from Pedersen et al. (2017) that the body
belongs to the Asuk Mb.

In the upper part of the drill hole, the Q values are slightly lower than 1.0 such that predominantly slightly positive anomalies are expected at shallow depths, even if the body is formed during a period of phase reversal. This is in agreement with the magnetic survey data, where both positive and negative short-wavelength anomalies are observed indicating that in some areas the induced and other areas the remanent magnetization dominate in the shallow part of the body. In the 3D
inversion results such a distinction of two layers having clearly different magnetic properties cannot be observed. In contrast, a rather complex pattern of magnetization varying both in magnitude and direction suggests a more complex distribution of material with magnetic properties (as magnetite, native iron and sulphides), However, since magnetic inversions have a very limited resolution (even if the inversion is constrained), we cannot exclude that the magmatic body consists of two distinct flows with significantly different magnetic properties across its whole extent or in major parts.

Although the remanent magnetization seems to be predominantly oriented into a positive z-direction (the Earth's magnetic field has a large component in the downward z-direction), inversion results do not necessarily support that its direction is uniform and always parallel to the palaeomagnetic field direction. So, the obtained range in magnetic susceptibility equivalences (from 0.1 up 0.15 SI) in the first constrained inversion run, where only the sediments were considered as non-magnetic, are in the same range of magnetizations as observed in the borehole FP94-4-5 (Fig. 10d, susceptibilities of 0.12 SI
and remanent magnetizations of 15.2 A/m). However, the values obtained in the second inversion run, where also the overall magnetization is assumed to be aligned about (parallel and) antiparallel to the (Earth magnetic field and) paleomagnetic field are unreasonably high (up to 0.4–0.6 SI; additional inversion runs in Appendix C) to represent the values from the petrophysical measurements. In addition, alignments of some larger trends in NS to NNS–SSE (Fig. 7 d–i) directions, as observed in the first inversion run, appear geologically more reasonable than the artificially appearing patterns with many
small-scaled features observed in the second run (Fig. C1). Accordingly, we assume that the results of the second inversion run are less meaningful than those of the previous run, although the target misfit was reached in the second run, making it a solution theoretically. This means that it is likely that the magnetization direction is a complex distribution and not only aligned along the paleomagnetic field and the current Earth's magnetic field.

Two aspects could explain varying directions of the magnetic remanence. In case that native iron measurably contributes to
the magnetic behaviour and a major component of its remanence has a viscous magnetic character, this could impact the overall direction of the remanence. It can be imagined that its magnetization direction changes after rock formation and accordingly deviates from the direction of the paleomagnetic field. Rotational components of the landslide and other mass movements change remanence directions in a complex manner; particularly in the unstable areas in the south. Because the magmatic body was presumably not impacted by major metamorphism or upheating events after its formation in the
Palaeocene (63 Ma), it is quite unlikely that the rocks reached Curie temperatures afterwards. As such, we see thermal demagnetization as unlikely reason for the complex magnetization directions.





**Figure 13.** Integrated interpretation. (a) RGB-composite plot from PlanetScope images (Planet Team, 2021) that are merged with semi-transparent TPI from the eBee DSM to increase image contrast. Landslide features are marked as dashed lines with thin lines for larger blocks and thick lines for the main landslide scarp. Talus below the Inussuk and block fields are the source for numerous boulders in the whole area. Mapped sediments reach up till the foot of the Inussuk cliff. (b) Overview maps illustrate area in relation to the different data,





b1: ArcticDEM at 2 m resolution with interpreted blocks, b2: PlanetScope greyscale mosaic with ground truth locations (spectroscopy and magnetic susceptibility) b3: Sketch showing the assumed extent of the target magmatic body (only the part covered with UAS data are shown). (c) Two schematic cross-sections in west-east direction are shown together with the magnetic anomaly and the analytical signal
from the UAS-borne magnetic, and iron ratios extracted from the drone-borne multi-spectral. The locations of these cross-sections are sketched in (a). (d–e) Coast-side views onto the magmatic outcrops in the (d) northern and (e) southern part of the study area.

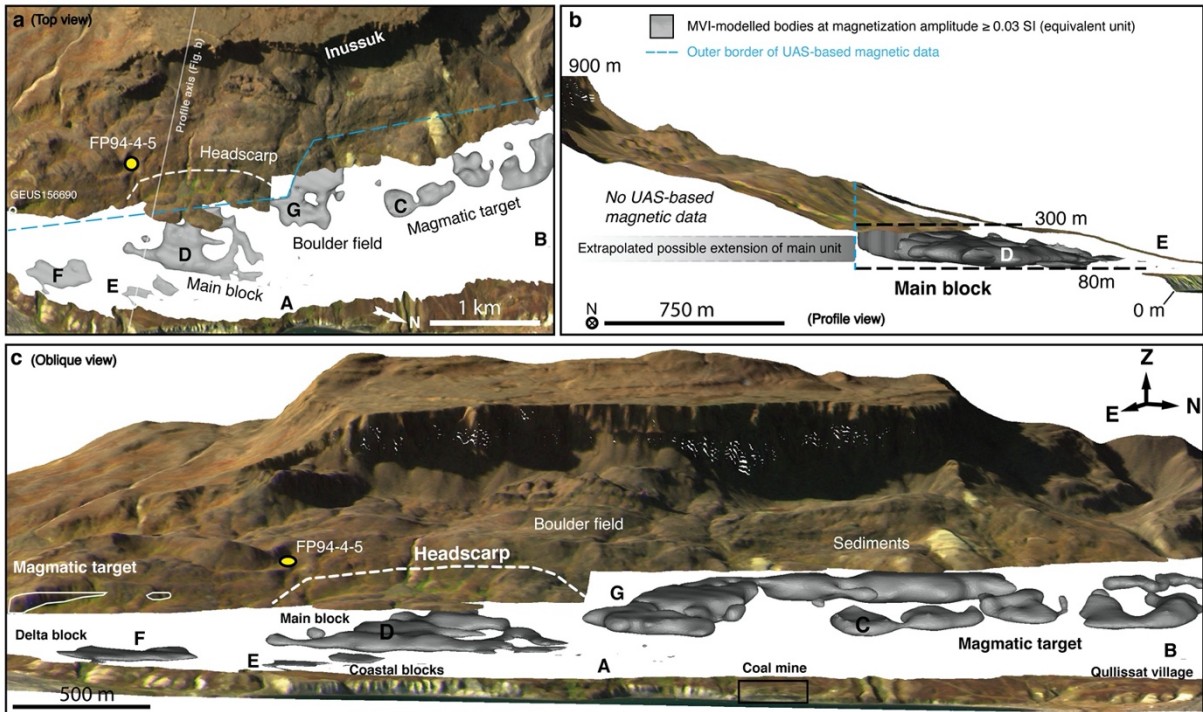

**Figure 14.** Combined plots of surface topography (PlanetScope mosaic fused with ArcticDEM) and larger magnetization amplitudes (≥ 0.03 SI) in the subsurface from the MVI model: (a) Top view of survey area. (b) Side view cross-section facing north. Dashed lines
indicate the maximum extension of the intrusive body. (c) Oblique view from the east. In all three views the surface model is removed for areas where the magmatic body is present at the surface (compare Figs. 12a, 13a). A possible extension of the target unit is sketched next to the main block, anomaly D.

## 5 Conclusion and Outlook

Our results demonstrate that an integrated workflow of UAS-based optical remote sensing and magnetic data with
supplemental measurements from the ground and drillholes is advantageous to precisely map and describe a mineralized magmatic body at depth and to improve the targeting of potential Ni-Cu-Co-PGE mineralization. Although the body is mainly hidden at the surface, we obtained significantly more detail than what was possible from all formerly acquired legacy data. Thus, we conclude the following:

(1)    The lateral extent of the magmatic body could be precisely determined from the distribution of the magnetic
anomalies and results from a constrained 3D magnetic inversion, where the upper and lower boundaries of the body were constrained from observations in the DSM and boreholes. It allowed us to develop a realistic three-





dimensional model of the body that agrees with and augments former interpretations (Pedersen et al., 2017). It proposes that the body has a flat-lying tabular shape and extends across large parts of the southwestern survey area.

(2)   The shapes and arrangement of the shallow magnetic anomalies indicate that larger blocks of the magmatic body were displaced by landslide movements. By identifying landslide features in the DSM and relating them to detailed magnetic anomaly patterns, it was possible to determine a precise location and extent of these blocks. Because major parts of the body do not outcrop, this would have been not possible from surface observations alone.

   (3)   The presence of negative long-wavelength magnetic anomalies indicates a significant remanence that proposes a reversed Earth magnetic field during its formation. This is in agreement with interpretation that the magmatic unit

was emplaced during the pole reversal associated with magnetochron C26r and is part of the Asuk Mb of the Vaigat Fm (Pedersen et al., 2017). A strong remanent magnetization component is confirmed by the petrophysical measurements from cores of borehole FB94-4-5 (Königsberger ratio up to 2.7).

   (4)   An unexpected observation from the MVI results is that the remanent magnetization direction in the body does not seem to be uniformly oriented parallel to the paleomagnetic field. The reasons for this are not understood yet.

Explanations could be potential rotational movements (e.g., from slope instabilities) after the Curie temperature was reached, or uncommon magnetic behaviour of native iron.

An abrupt change of the petrophysical measurements in borehole FB94-4-5 indicates that the magmatic body is split in an upper and lower part with distinctly different physical properties and presumably varying compositions. The most plausible explanation would be two separate magmatic events, i.e., lava flows, which supports a hypothesis that the bodies were

formed extrusively. Therefore, a detailed SEM mineralogical study has been initiated, but too few samples are systematically investigated in this manuscript to draw any quantitative conclusions of compositional variations. A combined quantitative mineralogical and petrophysical analysis at microscale is advised to clearly identify the minerals responsible for the magnetic anomalies. Native iron, pyrrhotite and magnetite are observed in hand samples and core pieces. At this stage we only conclude that magnetite is generally dominating the magnetization. Thus, our knowledge on the contribution of the

magnetic properties caused by pyrrhotite and native iron remains incomplete. In this context, it is both for exploration and in academic interest to investigate how native iron-related magnetization and its resulting anomalies are expressed. Native iron is geologically rare and its magnetic response in field practice is barely investigated.

Experiments to determine magnetic properties with increasing temperature and alternating fields will help in this context, e.g., as illustrated by Austin and Crawford (2019) on Ni-Cu-PGE mineralization in Australia. From such measurements, it is

possible to identify rock forming and environmental effects, metamorphic overprinting and a better geologic understanding of how such magmatic suites form. We have decided to postpone these measurements at this stage, since the core orientations are unknown and accordingly much of the obtained directional information would be spurious. Oriented core samples would need to be gathered first.

A far-reaching goal is a better understanding of sources (native iron, magnetite, graphite and pyrrhotite) that cause the

observed anomalies in different types of geophysical data, not exclusively magnetics. This requires incorporating, e.g.,





electrical properties from samples and results from airborne EM and MT measurements, which might have characteristic responses for graphite, pyrrhotite and presumably native iron accumulations. In this context, one of the existing MT profiles in the Qullissat area shows a very pronounced high-conductive anomaly immediately outside the western border of the magnetic UAS survey at a depth that correlates with the bottom of the magmatic body (pers. communication with Blue Jay

Mining Plc). There are different hypotheses about its origin and plans for a new borehole at this location. Finally, we recommend employing integrated multi-sensor UAS-based survey methods for geologic targeting to identify mineral groups and to improve distinguishing iron-bearing mineralogy.

**Appendix A: Flight control and processing of UAS-based magnetic data**

Flight performance is controlled by livelink software via telemetry link. A GSM based tracker device allows locating the

UAS in case of lost telemetry signal. A magnetic and barometric base station, which is placed near to the mobile telemetry/control station, measures the time variant part of the magnetic field (three-component fluxgate magnetometer) and the barometric pressure.

In the post-processing of the data, the magnetic field (and optionally the barometric altitude) measured by the moving UAS platform are corrected by the variations measured at the base station. For these surveys, the dedicated base station

magnetometer was not available, and a replacement UAS magnetometer was used for the base. The basic data processing (Table A1) utilized the RadaiPros software (version 2.0, Radai Oy, Oulu, Finland). After these general processing steps, 1) data were checked visually and invalid or unnecessary points (e.g., spikes and flights between the home base and the survey area) were removed, 2) base station correction was applied to the magnetic total field data, 3) separate flights were combined and data outside of the survey area were taken out (with 25 m margins), 4) a low-pass filter with a cut-off wavelength of ~20

metres was applied, (5) a ELM was conducted and finally (6) the IGRF core field was removed.

In the ELM method, first a deterministic inversion is used to find a simple susceptibility model, whose synthetic response fits the measured magnetic total field. In a second step, the obtained susceptibility model is used to compute the total magnetic intensity at a constant elevation level on an even grid. Here, the susceptibility model is composed of a single layer of three-dimensional magnetized cells.

The ELM method reduces the effects of varying flight altitude and uneven sampling of the data points, high frequency noise and artifacts provided that their wavelength is short compared to the size of the elements used in the layer model. ELM is used also to level the datasets and make the heading correction. The ELM was applied to the fully pre-processed total field data. Horizontal size of the elements was 40 m × 40 m and their vertical height was 111 m. Depth to the top of the model was 2 m and the top of the model followed the terrain topography. Total number of elements was 148 × 64 = 9472, and the

total number of (decimated) data points was 12531 (15.10 %).

Table A1: Processing steps of UAS fixed-wing magnetics.



| 1 | Remove dummy values |
|---|---|
| 2 | Computation of barometric height (relative altitude) from pressure data. |
| 3 | Computation of rectangular X and Y map coordinates (UTM 22W). |
| 4 | Computation of running profile distance coordinates and azimuth/heading angles, |
| 5 | Application of fluxgate calibration parameters (derived from a separate calibration measurement). |
| 6 | Computation of the raw and (orientation) corrected magnetic total field. |

### Appendix B: Multispectral surveys

We used Agisoft Metashape (version 1.6, Agisoft, St. Petersburg, Russia) to process multispectral data, create digital surface models as raster and 3D point clouds, and multispectral orthomosaics from each flight area, and the complete flight area in one image. A total number of 11 eBee MSI flights in four sub-areas were conducted totalling 308 line-kilometres. The flight lines were set parallel to the regional slope with terrain following and with a forward- and sideward-overlap of 80 % and 60 %, respectively.

For spectral calibration, three images with different image acquisition settings (i.e., different integration time to adjust for varying illumination strengths) were taken from a ground reference target (Airnov$^{TM}$ VIS-NIR grayscale panel). These calibration images are automatically detected in Agisoft Metashape or selected manually. Our selected settings in the image alignment step are: alignment accuracy in high; pair selection with reference; adaptive camera model fitting is activated. After each step, the points were filtered with gradual selection repeatedly, and the option "optimize cameras" was applied

each time to reduced spatial errors (see 3D remote sensing lab at SLU Umeå, Sweden, http://www.rslab.se/agisoft-photoscan-pro, last accessed 04.03.2021). After the correction steps were applied, the sparse points were used to create a dense point cloud, using the "high quality" settings in the creation dialogue. From the dense cloud, we exported the elevation model and the 3D point cloud (reference standard WGS84 UTM 22N, EPSG 32622).

### Appendix C: Magnetic vector inversion with constrained magnetization directions

We assume that the remanent magnetization in the magmatic body has roughly the direction of the paleomagnetic field, when the body formed, since no tectonic processes with major rotations have taken place after the formation of the basalt. However, this must be considered carefully, since the central part of the investigation area has been affected by slope instabilities and rotational movements might be associated with them.

Referring to Riisager and Abrahamson (1999, 2000), the paleomagnetic field direction of the Asuk Mb can be estimated with

an inclination and declination of -80.7° and 228.1°, respectively. The nowadays Earth magnetic field direction (IGRF) at





Qullissat had during the field campaign an inclination of 81° and a declination of -31.5° resulting in an angle of ~168.3° between the two fields.

Accordingly, we assumed that the induced and the remanent magnetization are approximately parallel or antiparallel in most locations. We considered these main magnetization directions in the magmatic units by keeping the components
perpendicular to the Earth magnetic field small. This was achieved by modifying the weights $w_{o\ (Basalts)}$ of $\phi_{M,Ref}$ and having non-zero weights of 0.125 in the direction perpendicular to the current Earth magnetic field such that zero-weights only remained in the direction of the Earth magnetic field for the basalts and the magmatic body. Since the weights had to be expressed in the coordinate system used for the model (x = East, y = North, z = height), a rotation was applied onto the weighting factors resulting in $w_{o,p(Basalts)}$ = (0.124, 0.123, 0.0196). Rather small weights were selected with 0.125, since
there is a higher uncertainty with the assumed magnetization direction.





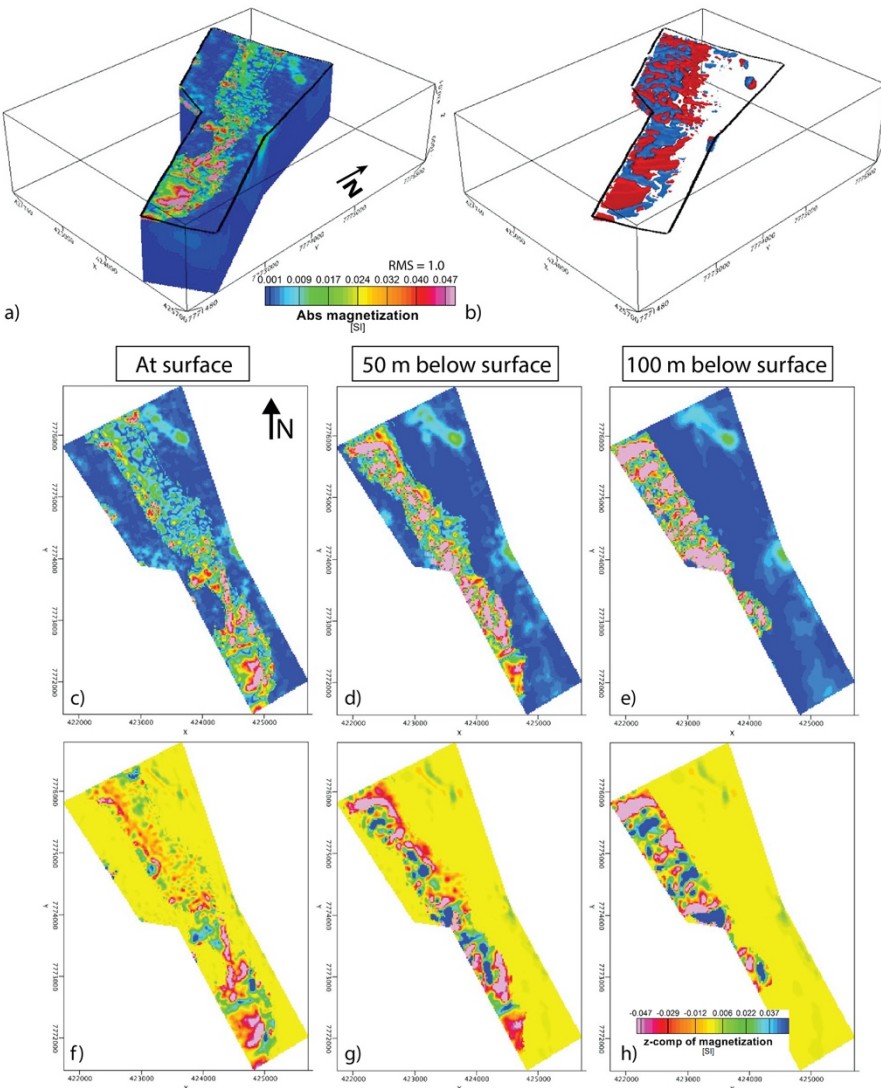

**Figure C1:** Results from the inversion model, where in addition the direction of the magnetization in the magmatic units is constrained towards the direction of the regional magnetic field. (a) The final magnetization distribution is presented as absolute values of the magnetization vectors. (b) Only cells with absolute magnetization values > 0.01 SI are shown as iso-surfaces. Blue and red colors are associated with locations, where the z-component of the magnetization points skyward, out of the ground (z-component is positive) and into the ground (z-component is negative), respectively. The remaining figures show the magnetization along the surface (c and f), and at depths of 50 m (d, g) and 100 m (e, h) below the surface, respectively. In (c–e) and (f–h), the absolute value and the z-components of the magnetization are presented, respectively.

Results of this test are presented in Fig. C1. As for the former inversion test, in which only the sediment units were constrained, higher magnetization values are accumulated in cells associated with the magmatic units and both positive and negative anomalies appear. The only anomalies located outside of this volume are again the anomalies A and B. In contrast to the former inversion run, the x- and y-components have rather small values within areas assigned to the magmatic units



(e.g., Fig. C) such that almost the whole magnetization is associated with the z-component. In addition, the absolute magnetization values are generally higher, but are usually not > 0.4 SI (maximum value is 0.6 SI), and general orientations in
a N-S to NNW-SSE direction are less pronounced (Fig. C1) than in the former test but anomalies are more scattered and generally more small-scaled.

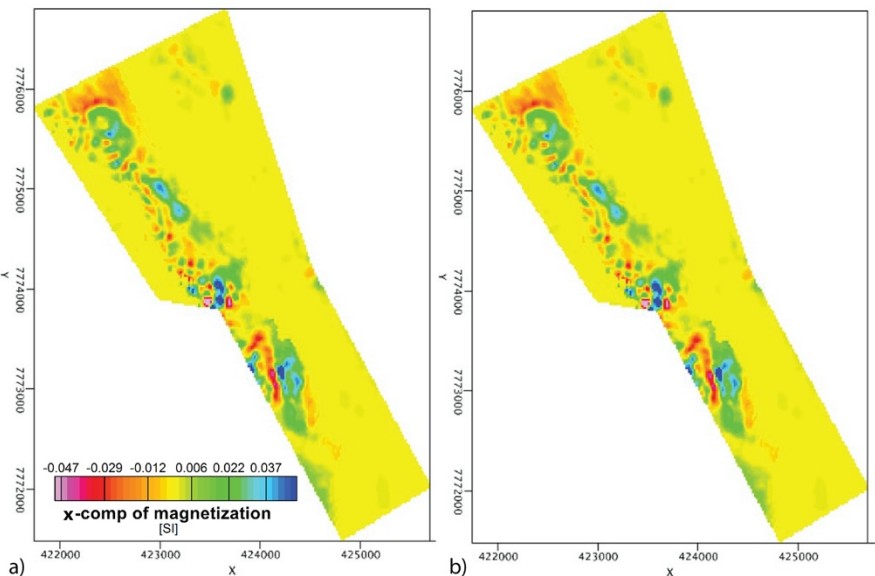

**Figure C2.** X-components of the magnetization 50 m below the surface are shown for the two constrained inversion tests. In (a) cells associated with sediment units were constrained towards a non-magnetic reference model (see Figs. 6 and 7), but in (b) in addition the
direction of the magnetization in the magmatic units is constrained towards the direction of the regional magnetic field (see Fig. C1).

*Data availability*. Original data of the study is available upon provision of a reasonable request.

*Author contributions*. RJ, RZ, BH, HS and UK designed the survey and conducted field work. RJ wrote the draft with
support of BH, EV, SL, RZ and MK. RJ constructed Figs. 1–4, 8–10, 12–14 and integrated all data. BH conducted the inversion and built Figs. 5–7, C1, C2. SL provided SEM analysis and Fig. 11. RZ acquired optical UAS data. RJ processed and analysed optical UAS and surface sample data, HS processed ground magnetics. MP processed magnetic UAS data. All authors took part in the result discussion, interpretation and review. BH and RG supervised the project and acquired funding.

*Conflict of interest*. The authors declare that they have no conflict of interest.

*Special issue statement*. This article is part of the special issue "State of the art in mineral exploration". It is a result of the EGU General Assembly 2020, 3–8 May 2020.





*Acknowledgements.* We thank Blue Jay Mining Plc for their support during the preparation and conduction of the field work. In particular, we thank Bo Møller Stensgaard and Hans Jensen for extensive logistic support. We thank Radai Oy, especially Arto Karinen, Lauri Maalismaa and Ari Saartenoja for the UAS-based magnetic measurements. We thank GEUS for preparation of the field campaign. Lotte Larsen, Asger Pedersen and Ethan Barnes are gratefully thanked for the fruitful geologic discussions. We thank Satu Mertanen from GTK petrophysical labs, and the Erzlabor at HZDR-HIF with Kai

Bachmann are thanked for sample preparations and measurements. MULSEDRO's field campaign was conducted under scientific survey licence (VU-00158-2019) within mineral exploration license MEL 2018-16 of Blue Jay Mining Plc.

        *Funding.* This research in the EIT project MULSEDRO was funded by EIT Raw Materials (project ID 16193) of the European Union organization EIT. Blue Jay Mining Plc supported the SEM analysis.

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
