# Peer review of "Drone-based magnetic and multispectral surveys to develop a 3D"

_Solid Earth, 2021_

## Author Comment (AC1)

We thank the reviewer for his careful reading and great suggestions to improve the manuscript and express our gratitude to the positive evaluation.

**Dear Authors,**

**The manuscript is informative and useful to mining-related sectors. The research combines different methods and provides detailed analysis. It can be accepted for publication after some modifications.**

**In L27 and L81, data are plural.**

Changes are made as recommended.

**Please define abbreviations before using them, such as UAS and PGE.**

UAS was already introduced in line 26. We have added the meaning of PGE (Platinum Group Elements). We thank the reviewer for this remark and we checked further abbreviations.

**The figures are not cited in the order. Readers have to jump about to find the figures regarding the text. I suggest better organizing the figures. Maybe removing some subfigures is a solution. More comments on the figures are below.**

The figure order has been corrected by changing their appearances in the body text, and by removal of superfluous mentioning. In addition, we have changed the order of figures 13 and 14.

**Why the multispectral survey area is not overlapping the magnetic survey area?**

The main reason why the areas of the magnetic survey and the multispectral surveys do not fully overlap is that we had technical problems with the magnetic drone system in the first half of the field campaign and we had not the time to fly the entire area of interest with the magnetic drone. In addition, we considered it as too risky to survey areas very close to the very steep and hundreds of meters high Inussuk cliff and, here, a larger safety margin for flight altitudes may be considered.

**In L326, most of the anomalies are trending in an NW-SE direction? I don't see it in the figure.**

We agree with the reviewer that we may have overinterpreted the directional trend of the anomalies. We have  modified the sentence and it is not mentioned any more that the anomalies are trending into a specific direction. We want to point out, that this direction trend is also influenced by the main flight line direction, forced by the necessity to fly with high safety for the equipment.

**Anomalies in Fig. 4 are not well labeled. A seems in the sea.**
We modified and updated the labelling of the anomalies in Fig. 4 to increase the readability.

**Since you didn't indicate the anomalies with arrows, you would better keep the marks at the same location in all the subfigures.**

The labels of the identified anomalies are placed now at the exact same locations in all subfigures of Figure.4. In addition, we have used arrows for indicating the locations of anomaly A and B. See also the answer above.

**F seems moving quite a bit and B and C are also shifted a bit in the different subfigures. I also suggest marking the anomalies in Figs 12 and 13. Since the N is oriented differently in those figures. It is not easy to follow what you refer to in the discussion.**

As recommended by the reviewer, we have added the labelling of the anomalies also in the overview figures 12, 13 and 14 to better link the magnetic anomalies to other observations. .

**In L394, 'a constant altitude of 40 m were used as input' is written. However, in the early text, it says that the mean of the latitude is 70 m. Why do you use 40 m here?**

To calculate the data response onto a flight height of 40 m, which is slightly lower than the resulting mean flight height of 70 m, is not a major concern for the quality of the resulting residual magnetic anomaly grid as long as no major high frequency noise is introduced. We have used a dense flight line pattern (line spacing: 40 m; inline sampling: ~2.6 m) required for such a step and have carefully inspected the final grid but have observed no such kind of artefacts.

It is an interesting question from the reviewer, how such a change in the flight height of the modelled input data impacts the inversion results. Therefore, we have repeated our constrained inversion for an ELM input data grid of 70 m (using otherwise exactly the same parameters).

One may expect that the resulting magnetization anomalies are moved to a slightly deeper depth level for this inversion setup, since the sensitivities at very shallow depths are relatively lowered. However, an implemented depth weighting scheme in the inversion prevents this, and we observe that higher magnetizations accumulate in the same depth range for both inversion tests (see figure below).

Since also the shapes of the resulting magnetization distribution within the body are very similar for both tests, we do not see any major concerns to present the results from a test, where the used input data in the inversion are calculated at an altitude of 40 m.

[Figure]

*Fig. Constrained inversion results using as input ELM calculated data at an altitude of 40 m (a and c) and 70 m (b and d) above the surface topography. Only cells with absolute magnetization values > 0.01 SI are shown as iso-surfaces. Blue and red colours are associated with locations where the z-component of the magnetization points out of the ground (z-component is positive) and into the ground (negative z-component), respectively.*

**The reference model generated for the MVI is not convincing. The base/thickness of the magmatic body is constrained by a single borehole. Magnetic inversion is prone to push the anomalies upwards to the surface. Besides, if the base is varying in 3D, the magnetic inversion may not resolve it. If you use a thicker magmatic in the reference model, the inversion will eventually fit the data too. But your results will be different. Therefore, more information at depth is needed. You have cited the results from MT data and airborne EM data. Why not use the subtracted resistivity models from the papers to better constrain the reference model, especially, at the depth? Those profiles are in 2D and probably overlapping with your survey area, it should be useful. Please show the data fits too.**

We agree with the reviewer's understanding that the maximum available information should be used to make an as reliable as possible estimate of the base of the magmatic body. We mentioned in the introduction that "*Legacy data, which were all available to the authors of this paper, include airborne magnetic and active electromagnetic (EM) data …*". This sentence indeed gave the impression that use of the EM data could be of help to refine the estimate for the base of the magmatic body, which is, however, not really the case:

- Inspired by this comment from the reviewer, we have tried to make an inversion of the airborne EM data. However, we were unable to invert the (legacy) GeoTEM data from the 90s due its unknown data normalization method and lacking system parameters about the shape of time varying current signature in the primary coil. Accordingly, it is hardly possible to extract reliable information with depth.

However, the decay constant in the off-time window can be used to qualitatively map conductivity anomalies that can be associated with accumulations of graphite, sulphides and/or native iron. These conductivity anomalies were used by a former license holder to define the drilling locations.
(Please note that a package with the EM data can be downloaded from the Greenland Portal; *www.greenmin.gl*).

- Magnetotelluric (MT) is an excellent method to precisely determine the location of conductors, but it is known that the method has low resolution power to describe resistors. Since a magmatic body as e.g. basalts are typically resistive (provided that not very large amounts of sulphides, native iron and graphite are accumulated in the body, which would make it more conductive again), a thickness estimate from a MT inversion is also imprecise if conductive sediments are located underneath.
One of the co-authors has actually published a paper (Heincke et al., 2017; Journal of Applied Geophysics), where it was investigated to extent a basalt layer having sediment units above and underneath can be resolved (1) with a single MT inversion and (2) by a combination with other geophysical data (gravity and seismic topography) in a joint inversion. These results demonstrate that the single MT inversion both underestimates  resistivity values and  thickness of the basalt due to the impact of conductive sediments above and below. For our study, this means that independently inverted MT data can give information about the existence of sediments underneath, but no exact estimate about the thickness of the magmatic body.
A combination of magnetic data and MT in a joint inversion approach could solve this uncertainty about the thickness to some extent. Such a joint inversion strategy would, however, be not straightforward since reliable petrophysical relationships between magnetic and conductive properties are probably hard to establish and structural linkage (e.g. by a cross-gradient constraint) provides only little improvements if methods with low resolution as e.g., magnetic are involved (see e.g. Lelievre et al., 2012, Geophysics). A solution could be to run a joint inversion with other, more advanced constraints as the very recently proposed constraint that uses mutual information (Moorkamp, 2021, First International Meeting for Applied Geoscience & Energy Expanded Abstracts). However, these are brand-new technical developments, whose applicability has to be proven for such exploration studies first.. It is above the scope of this paper to apply them here, where we primarily focus on our drone data. But such approaches are interesting to improve such exploration models in future.
(Please note that a package with the MT data can be downloaded from the Greenland Portal; *www.greenmin.gl*).

Nonetheless, these EM data are of much value for mineral exploration at Qullissat because of a different reason. They allow determining conductive anomalies that directly pinpoint areas that can be associated with accumulations of native iron, graphite and/or sulfides.
We had therefore some discussion to add EM results to the paper or not, but decided not to do it because
1. the manuscript had already a significant length (and discussion of conductivity anomaly distribution would extend it by several pages) and
2. the current license holder (BlueJay Mining) requested us not to present the distribution of anomalies observed in the MT sections.

We have done now following changes in the manuscript:

(1) We modified this misleading sentence in the introduction that gave the impression that all EM data are fully available and of much use for defining the thickness of the basalt.

(2) We modified at several places the text of the result section (1) to indicate the limited resolution of the magnetic method and (2) that the lower boundary estimate has a larger uncertainty.

(3) We have added a map (see Figure 3b) that shows the decay constant in the off-time window from the airborne EM data. This map does not provide any further insight into the thickness of the magnetic body, but it shows some of the conductivity anomalies that were considered as targets for further exploration by the former license holder Falconbridge Limited in the Qullissat area.

(4) We see different types of measurements that could generally improve the exploration model - and in particular the thickness estimate of the magmatic body - in future, and we give some recommendations in the Conclusions & Outlook section, what type of data could be acquired. We believe that particularly a new TDEM survey should be flown with a modern higher-resolution system (e.g. VTEM or SkyTEM). Results from such a survey should help to provide the missing constraining information about the thickness of the body.

In this context, we would like to emphasize that we, the authors of the paper, were in charge of the drone-based data (and related ground truthing data) acquired in a recent European Union funded project. In addition, we added the otherwise missing petrophysical and mineralogical measurements from the borehole, but we had no impact on the quality and type of legacy data acquired by exploration companies in the past (15 to 25 years ago).

The mineral exploration company BlueJay Mining has some activities and plans to fill these gaps with new data. E.g. a new borehole cutting the base is discussed and an academic consortium (D-REX consortium) will conduct a 3-D magnetotelluric survey on Disko this summer (2022).

References:

Heincke, B., Jegen, M., Moorkamp, M., Hobbs, R. W., & Chen, J. (2017). An adaptive coupling strategy for joint inversions that use petrophysical information as constraints. Journal of Applied Geophysics, 136, 279-297.

Lelièvre, P. G., Farquharson, C. G., & Hurich, C. A. (2012). Joint inversion of seismic traveltimes and gravity data on unstructured grids with application to mineral exploration. Geophysics, 77(1), K1-K15.

Moorkamp, M., (2021). Joint inversion of gravity and magnetotelluric data from the Ernest-Henry IOCG deposit with a variation of information constraint. Conference Proceedings of First International Meeting for Applied Geoscience & Energy. DOI:10.1190/segam2021-3582000.1

**Concerning " Magnetic inversion is prone to push the anomalies upwards to the surface."**

We have forgotten to mention in the description of the inversion that a depth weighting scheme is implemented in the inversion to avoid that sensitivities dominate at shallow depths and push magnetization anomalies upward.

We have added a sentence about the depth weighting scheme in the section describing the MVI inversion settings..

**Concerning "Please show the data fits:**

For all inversions, an error weighted RMS data misfit of 1.0 was reached and we added the formula of the RMS misfit to the section describing the MVI inversions.

Below, we have, moreover, added a figure that shows the observed residual magnetic anomaly (input data in the inversion) and the corresponding calculated magnetic response (predicted data of final model).

Because the manuscript is already long, we believe it is sufficient to only present the overall RMS misfit in a formula and it is not needed to present further figures..

[Figure]

Observed magnetic anomaly

Predicted magnetic anomaly

| 359.0 |
| 317.9 |
| 276.9 |
| 235.9 |
| 194.9 |
| 153.8 |
| 112.8 |
| 71.8 |
| 30.8 |
| -10.3 |
| -51.3 |
| -92.3 |
| -133.3 |
| -174.4 |
| -215.4 |
| -256.4 |
| -297.4 |
| -338.5 |
| -379.5 |

**Observed data**
[nT]

**Calculated data**
[nT]

*The figure shows the observed data response used as input for the inversion (left) and the resulting predicted data response from the inversion (right) from the constrained inversion model. The resulting error weighted RMS is 1.0 assuming a constant data error of 5 nT.*

**In Figs 10 and 11, the analysis of a sample at the transition depth (~127m) is not shown. All the samples of cores are from the lower basalts. What do the ones from upper basalts (at 58-127 m depth) look like? What are the fillings among different minerals? Do they play a role to show different susceptibility?**

That is a valid observation from the reviewer. For Fig. 11, we have improved the figure annotation to clarify that the upper image (Fig.11a) and the lower image (Fig. 11b) show a thin section from the upper and lower part of the borehole , respectively.
The fillings among the minerals are mostly amorphous-glassy phases consisting of Si-Al-K-O and contain no measurable iron. Those phases should not have any influence on the measured susceptibility.
The labels for Fig. 10e have been improved and additionally include downhole depth next to altitude above sea level (asl) for clarity. The formerly shown meter asl values were intended to create a comparability with the shown elevation isolines in respect to the magmatic target body. In fact, representative core photos from the upper basalts were shown in Figures 11c-e, but the labelling was misleading as it used the altitude above sea level.

The mineral composition does not change fundamentally from the depth, where the thin section is presented (~87m downhole), down to the transition depths of 127 m.

**In L585, the main target body is about 7.5 km2, however, the magnetic survey is about 6.8 km2. How do you fill the gap? Also, some parts of the magnetic anomalies may be caused by the drifts from the main body due to landslides.**

Figure 13 has been adapted and the value in the former Line 585 was modified. The originally estimated area included an extrapolation outside of the UAS-based magnetic survey area and also included information from boreholes and mapped outcrops in the regional geologic map. Now, the value has been re-estimated based on the modelled target body and confined inside the magnetic survey area. We estimate a surface area of 4.5 km$^2$ excluding the large detached blocks.

**It is better not to say the constrained information agrees with the inversion result. The two should not conflict with each other.**
We agree with the reviewer and have modified the sentence as suggested.

**The magnetic data don't well resolve the structure at the depth. And the depth of investigation from the inversion is limited, so the statement in In L604-606 is not convincing. Comparing the results with MT and airborne EM results would be useful.**

- We agree with the reviewer that magnetic as a low resolution method in combination with a relatively narrow survey make it very difficult to reliably extract any deeper feature information out of the data. We therefore changed the sentences in the lines L604-606 to: „Because the investigation depth to resolve structures is very much limited for this local relatively narrow magnetic survey, it is hard to evaluate if any major magnetic material e.g., from intrusions is located underneath the body. In any case, the 3-D inversion results show that an approximately horizontal magmatic body with reasonable magnetization can explain the full data response and no deeper-seated structure e.g., associated with a feeder structure need to be added to fit e.g., long wavelength trends."
- We have added the decay constant determined from the legacy airborne electromagnetics data as a new subfigure in Figure 4. Additionally, we marked the two conductivity anomalies that were described in Olshefsky (1992) and Olshefsky & Jerome, (1993). Those two EM anomalies lead to the decision for exploration drilling at the time.
- Otherwise, we would like to refer to our response to a former reviewer comment.

**Hopefully, the comments are somehow useful. Cheers**

We are grateful for the critical comments from the reviewer, who makes us aware of a number of critical aspects that needed further consideration.

---

## Author Comment (AC2)

We would like to thank this referee for his comments. Please find our responses below.

**REF2**

**Dear Authors,**

**Overall, a comprehensive and informative case study which combines a wide variety of remote sensing (legacy geophysics, UAS-based magnetics, ground magnetics, UAS-based multispectral) and laboratory measurements (petrophysical measurements, borehole logging, mineralogical SEM analyses) to further the exploration understanding of Ni-Cu-Co-PGE-Au mineralization at Qullissat, Disko Island, Greenland. The manuscript provides a detailed summary and combination of several state-of-the-art exploration methods and discussion on how these methods were used and combined to improve mineral exploration within challenging/remote terrains.**

**Listed are some minor comments that would help clarify and improve the manuscript. These comments are listed based on the page numbering, line numbering in SE-21-133.**

**(Page 8, Line 225) – Briefly state whether the UAS-based magnetic flight lines were draped over the topography model.**

During the early planning stage of many aeromagnetic surveys, field survey planners define that flight altitudes should describe draped surfaces above the ground (i.e. the plane does not lower the flight heights, when crossing canyons and narrow valleys). In contrast, a constant height above the surface topography is defined in our UAS survey, but this is automatically modified by the flight plan software for areas having steep topographies to avoid that the plane has large pitch angles. These modifications result in flight heights that are similar to the ones of draped surfaces (i.e. the flight height is also larger in areas with large topographies as in narrow valleys, see figure below), but the way they are determined are different.
We have added a sentence in the section about the data acquisition to clarify this.

[Figure]

*UAS flight elevation above the topography. Large-attitude outliers are present in areas of local hills and higher basalt pillars.*

**(Page 9, Line 230) - What was this 1-5 nT magnetic noise in the raw data mainly from… platform electromagnetic interference, sensor motion? Briefly state the reason for the magnetic noise…**

Since the data error is accumulated by various noise sources during the acquisition (e.g.electromagnetic noise from electronic components, servos and motors, long wavelength temperature drifts and inaccuracies from accelerations of the fluxgate coils by rapid plane movements and vibrations), but is partly reduced by the processing (e.g. the ELM is an inversion method that reduces the impact of various noise sources), it is to some extent challenging to define an exact error estimate.
We have adapted the respective lines and present an estimate for the overall error of ~5 nT, which appears reasonable for us and is also used as an error estimate in the inversion.

**(Page 10, Figure 2) – Make the outlines a different color than the data you present in Figure 2a. Both present light blue and purple colored data and figure outlines, which are challenging to differentiate. Check color contrasts and consistencies in other figures.**

We thank the reviewer for this observation. The color of the outlines in figure 2 have been adapted and have now higher contrast. Additionally, the box colors for the zoomed-in maps were updated.

**(Page 19, Figure 17) – In the three depth slides, the magmatic body shifts to the West and changes shape slightly with increasing depth (surface to 100 m below surface) as would be expected given the topography and horizontal nature of the formation. However, the magnetization anomalies associated to points A and B remain larger unchanged in shape, position, and magnetization strength. Why is this the case? If they are assumed to be blocks of the main sill that slide down the hillside (Page 27, Ln 610) or anthropogenic sources, would it be expected that the magnetization would remain unchanged and extend to depth? Maybe a brief sentence or comment clarifying or addressing these assumed point source anomalies and how they interact with the inversion models.**

Yes, the different appearance of anomalies is indeed linked to the different inversion regularizations used inside and outside the volume associated with the magnetic body. Inside the body, there is no constraint towards a reference model and, hence, there is a larger degree of freedom that the magnetization can vary. In contrast, the area outside of the body is constrained towards a zero magnetization model such that introduced anomalies tend to have smaller magnetization values. To fit observed data anomalies in this area, it is then required that the magnetizations are spread over larger volumes (and wider depth ranges). We have added two sentences explaining the appearance of anomalies A and B.